# Tipping point analysis helps identify sensor phenomena in humidity data

Valerie N. Livina[1], Kate Willett[2], and Stephanie Bell[1]

[1]National Physical Laboratory, Teddington, UK
[2] Met Office, Exeter, UK

**Correspondence:** Valerie N. Livina (valerie.livina@npl.co.uk)

**Abstract.** Humidity variables are important for monitoring climate. Unlike, for instance, temperature, they require data transformation to derive water vapour variables from observations. Hygrometer technologies have changed over the years and, in some cases, have been prone to sensor drift due to aging, condensation or contamination in service, requiring replacement. Analysis of these variables may provide rich insight into both instrumental and climate dynamics. We apply tipping point analysis to dew point and relative humidity values from hygrometers at 55 observing stations in the UK. Our results demonstrate these techniques, which are usually used for studying geophysical phenomena, are also potentially useful for identifying historic instrumental changes that may be undocumented or lack metadata.

## 1 Introduction

Studying climate variables requires complex measurements and transformations of data. During the period of a given continuous record, observations may be derived from successive instruments. Sensors may undergo multiple technical changes, from replacements of instruments to drifts and degradation. Detecting such changes, especially during earlier periods of deployment, which were not fully documented, is an important task that contributes to better interpretation of data that forms climate records — in particular, to enable homogenisation processes that identify and address inconsistent data (Menne and Williams, 2009; Peterson et al, 1998). Homogenisation, detection of changes, and recovery of missing metadata also contribute to generic FAIR data principles: Findable, Accessible, Interoperable, and Reusable. In particular, it is necessary to produce robust long-term analyses to assess climate change, that changepoint detection at monthly timescales is relatively well established (Reeves et al, 2007).

There are various different methods for homogenisation of monthly data (Domonkos et al, 2021; Venema et al, 2012) amd some benchmarking of daily data (Killick et al, 2022; Brugnara et al, 2023), while hourly homogenisation is still a missing link. As now we move into an era of high-frequency and global-coverage climate services, there is the need for robust daily and even hourly analyses to assess extreme events, their intensity and frequency (Trewin, 2013; Brugnara et al, 2023). This makes monitoring climate sensor networks and their quality particularly important.

It is known that during the course of observations in the past few decades, manual observations were replaced by automatic ones, and that many analogue instruments were replaced by digital ones at various stages of environmental observations. Some

of these changes were not fully documented, or the records are not now accessible. Some instrument types have exhibited significant drift during period of service in between checks, electronic hygrometers being one example. Furthermore, a station may have moved (yet its metadata identifies it as at the same location) or instruments re-situated within the station grounds, or the local environment may have changed (urbanisation, city to airport, agricultural practices, afforestation or deforestation).

Therefore, it is important to analyse environmental records using suitable data techniques that are able to detect such artefacts and distinguish them from the natural phenomena being observed.

In the climatological community, there are various techniques developed for detection of outliers, which are used for verification of forecasts and data assimilation, i.e., combination of observations with short-range forecasts to estimate the current Earth system state.

In a series of ECMWF (European Centre for Medium-Range Weather Forecasts) publications (Dahoui et al, 2014, 2017, 2020; Dahoui, 2023), there were developed soft and hard limits for detection of sudden changes and slow drifts in climate statistics. These techniques were designed for in-house use, for real-time operational data checks, in particular, regarding satellite data contamination. Some of such methods are based on monitoring changes in standard deviation (Dahoui et al, 2014), whereas others (Dahoui, 2023) utilise more complex algorithms that combine several ML/AI techniques, such as autoencoders, Long-Short Term Memory (LSTM) deep networks, and classifiers. This approach can lead to good results after training on a specific variable with long observations, but it may also suffer the artefacts of overfitting.

(Todling, 2009; Todling et al, 2022; Waller et al, 2015; Waller, 2021) reviewed several techniques based on the approach that is called observation-minus-background or observation-minus-forecast (the former used for uncertainty quantification, while the latter one used by climate centres for ensemble forecasts and data assimilation). The essence of the approach is estimation of error covariance, which then can be used for analysis of observational deviations (which is the topic of the current paper). However, in such methods, several assumptions are applied, such as absence of correlations between observation and background errors, and the analysis errors are expected to be related lineraly with observation and background errors. We note that such assumptions are not imposed when applying the technique proposed here.

ML/AI techniques has became popular in other areas of fault analysis, in particular, in engineering applications. These techniques range from the regression-based residual analysis, with information criteria and error estimates for quantification of performance, to recently developed neural-network-based anomaly detection (for example, convolutional autoencoders), isolation forests, and support vector machines (Ciaburro, 2014).

In many of such techniques, it is necessary to train a neural network or fit a model on a historical data with known (labelled) anomalies. When such data is not available, performance on poorly-trained data may be unsatisfactory (80% or lower).

Yet another approach that has been applied for fault detection is Bayesian network (Yang et al, 2022). The approach provides uncertainty quantification, which is an advantage, but there two major limitations. First, it is most suitable for a multivariate process, which means that the input of a single time series may be not sufficient. Second, Bayesian networks require control limits for anomaly detection, using likelihood indices, which is done using kernel density estimation, and this requires additional choice of parameters (bandwidth values in several functional spaces). We applied a few such techniques to Bingley dataset but the results were not satisfactory (not shown here).

Given the available techniques described above, still, there is a need for a technique that would be sufficiently simple, yet sensitive, and, without supervision or multiple parameter fitting, capable of detecting diverse changes in times series.

In this article, we study humidity records of several decades using tipping point analysis that is capable of detecting long-term natural phenomena, such as climate change, as well as abrupt changes of data patterns caused by technical artefacts. This is the first time that tipping point analysis is deployed for such a purpose, and we demonstrate its usability for detection of instrumental changes. If used in real-time, utilising a small window size for up-to-now available data (thus reducing uncertainty in timing the change), tipping point analysis may help identify drifting of sensors and, in principle, prevent prolonged recording of poor-quality data. If used for historic datasets, a change in the noise characteristics might help identify a change in technology (for example from psychrometer to electronic hygrometer) in cases where this might not otherwise be recorded or detectable. Improving such information can potentially improve the attribution of uncertainty in cases where only a worst-case uncertainty might otherwise be assigned.

## 2 Methodology

### 2.1 Overview of method

In the work reported here, the technique more conventionally used to identify tipping points of complex systems was applied to several time series of near-surface observations of air temperature and of humidity at UK locations. After potential change points were identified using the analysis, the observation records and their metadata were assessed to investigate whether the identified change points corresponded to significant events such as the change of an instrument or of a recording technique.

### 2.2 Tipping Point Analysis

After Poincaré's pioneering work on bifurcation theory (Poincare, 1892), in the 1960s and 1970s an intuitive idea of a bifurcation appeared in social sciences as the term 'tipping point' coined by sociologists. Malcolm Gladwell published a bestselling book on tipping points (Gladwell, 2000), where he expanded the approach to biophysical systems. In 2008, Lenton et al. published the seminal paper on tipping elements in the Earth system (Lenton et al, 2008), in which the main geophysical tipping elements were described. Lenton's work gave an onset to multiple publications in geophysics and paleoclimate, focussing on early-warning signals (EWS) of tipping points (Livina, 2023). Applications of tipping point analysis have been found in geophysics (Lenton et al, 2008; Livina et al, 2011, 2010, 2013), statistical physics (Vaz Martins et al, 2010), ecology (Dakos et al, 2012), structure health monitoring (Livina et al, 2014) and failure dynamics in semiconductors (Livina et al, 2020). Most of these studies were focussed on long-range changes in dynamical systems, such as climate change, or on engineering phenomena, such as failures of devices or installations. In this paper, we return to geophysical applications, but with the purpose of studying sensor conditions, which bridges the gap between such applications and broadens the scope of tipping point analysis. We also supplement the analysis with a Bayesian estimate of significance of changes, which we illustrate on humidity data.

We propose to use the tipping observed in fluctuations (changes in the patterns of a dataset noise, which are not necessarily in the form of a geophysical tipping point) to identify changes in the patterns of the instrumental data, attributable to changes of the instrument or the observing technique.

The basis of the EWS in tipping point analysis is monitoring changes of patterns in fluctuations using approximating stochastic models. Such models are powerful yet simple tools for modelling time series of real-world dynamical systems. Given a one-dimensional trajectory of a dynamical system (the recorded time series), the system dynamics can be modelled by the stochastic equation with state variable $z$ and time $t$:

$$\dot{z} = D(z,t) + S(z,t), \tag{1}$$

where $\dot{z}$ is the time derivative of the system variable $z(t)$, and $D$ and $S$ are deterministic and stochastic components, respectively. Component $D(z,t)$ may be stationary or dynamically changing (for instance, containing long-term or periodic trends or both).

The tipping-point analysis consists of the following three stages: 1) anticipating (pre-tipping, or EWS detection/analysis), 2) detecting (tipping), and 3) forecasting (post-tipping). For the purpose of the current analysis, we use only the first stage of the tipping point analysis: anticipation, or early-warning signal.

**Anticipating tipping points** (pre-tipping) is based on the effect of critical slowing down of the dynamics of the system prior to critical behaviour, i.e., increasing return time to equilibrium (Wissel, 1984). When a system state becomes unstable and starts a transition to another state, the response to small perturbations becomes slower. This "critical slowing down" can be detected as increasing autocorrelations (ACF) in the time series (Held and Kleinen, 2004). Alternatively, the short-range scaling exponent of Detrended Fluctuation Analysis (DFA) (Peng et al, 1994) may be monitored (Livina and Lenton, 2007). The lag-1 autocorrelation (Brockwell and Davis, 2016) is the value of the autocorrelation function at lag=1, the autocorrelation function being the average dependence between elements of time series at various steps, or lags. Lag-1 ACF is calculated in sliding windows of fixed length (conventionally, half of the series length) or variable length (for uncertainty estimation) along the time series, which produces a time series of an early-warning indicator. This indicator describes the structural dynamics of the time series. If the time series of the indicator remains flat and stable, the time series does not undergo a critical change (whether bifurcational or transitional).

If the indicator rises to a critical value of 1 (as one is the maximal value of normalised autocorrelation, provided no detrending or filtering is applied to the input data), this is a signature of early warning signal of critical behaviour. However, when any pre-processing is applied to the input data, this is likely to modify autocorrelation values, and indicator may not reach the maximal value 1. In this case, the important property of the indicator is its monotonic increasing trend. Such a monotonic trend of the EWS indicator can be estimated, for instance, using Kendall rank correlation.

Lag-1 autocorrelation is estimated by fitting an autoregressive process of order 1 (AR1), which is a common modelling tool in time series analysis:

$$z_{t+1} = cz_t + \sigma\eta_t \tag{2}$$

where $\eta_t$ is a Gaussian white noise process of unit variance, $\sigma$ is the noise level as a standard deviation, and $c = e^{-\kappa \Delta t}$ is the 'ACF-indicator' with $\kappa$ the decay rate of perturbations. Then, $c \to 1$ as $\kappa \to 0$ when a tipping point is approached. In addition, the DFA method utilises built-in detrending of a chosen polynomial order, which allows transitions and bifurcations to be distinguished in the EWS. These features can be identified by comparing several early-warning indicators, with and without detrending data in sliding windows (Livina et al, 2012). The paper of Livina and Lenton (2007) provided the first application of the DFA-based early-warning indicator to the paleotemperature record with detected transition using both ACF and DFA indicators.

**Detection** of a tipping point is performed using dynamical potential analysis. The technique detects a bifurcation in a time series and the time when it happens, which is illustrated in a potential plot mapping by colour the potential dynamics of the system (Livina et al, 2011, 2010). The technique of potential **forecasting** is based on dynamical propagation of the probability density function of the time series (Livina et al, 2013). In the current paper, we do not perform detection and forecasting stages of tipping point analysis and focus on anticipating tipping only EWS. Detection technique, which focuses on the number of system states, describes only a subset of the possible changes (genuined bifurcations), and this would be a limitation in the current study.

The theory of tipping point analysis is generic, and here it is useful for detection, for example, of such cases where a hygrometer progressively under-reads (locally indistinguishable from true value), then a site check establishes that the instrument is out of specification, and it is replaced with a new calibrated hygrometer. In data, such a change would look like a local step-change in the data that follows some drifting trend (this trend may have provided a short EWS). Another case could be that a wet– and dry–bulb hygrometer read a few times per day is replaced with an electronic sensor (that reads at the same temporal intervals), which is then replaced by an automated reading protocol logged at 1-minute intervals (these datasets might have different effective resolution, or different error (noise) characteristics). Such a change of sampling rate would immediately produce stronger auto-correlations in the data, which would be detectable using the lag-1 autocorrelation EWS technique. We are also interested to detect more complex instrument changes, such as station/instrument moves, instrument drifts, and various local environment changes.

## 2.3 Humidity measurements

Following (Willett et al, 2014), calculation of relative humidity in this work is based on several input and derived variables as follows:

Vapour pressure with respect to water $(e)$ in hectopascals
$$e = 6.1121 \cdot f_w \cdot \exp\left(\frac{\left(18.729 - \left(\frac{T_d}{227.3}\right)\right)T_d}{257.87 + T_d}\right)$$
$$f_w = 1 + 7 \times 10^{-4} + 3.46 \times 10^{-6} P_{\mathrm{mst}}$$

Vapour pressure with respect to ice $(e_{\mathrm{ice}})$ in hectopascals
$$e_{\mathrm{ice}} = 6.1121 \cdot f_i \cdot \exp\left(\frac{\left(23.036 - \left(\frac{T_d}{333.7}\right)\right)T_d}{279.82 + T_d}\right)$$
$$f_i = 1 + 3 \times 10^{-4} + 4.18 \times 10^{-6} P_{\mathrm{mst}}$$

$$(3)$$

Station pressure $(P_{\mathrm{mst}})$ in hectopascals
$$P_{\mathrm{mst}} = P_{\mathrm{msl}} \left(\frac{T}{T + 0.0065 Z}\right)^{5.625}$$

Relative humidity (RH) in percent relative humidity
$$RH = 100 \left(\frac{e}{e_s}\right),$$

where $T$ is station climatological monthly mean, $T_d$ is dew-point temperature (in Kelvins), $P_{\mathrm{msl}}$ is pressure at mean sea level, $Z$ is height in metres, $e_s$ is saturated vapour pressure.

## 3 Data

We consider humidity observations in the UK that span several decades from the Met Office Hadley Centre's Integrated Surface Dataset (HadISD), see (Dunn et al, 2012, 2016, 2019). This is a global hourly land surface dataset of core meteorological variables originally archived by NOAA's National Centers for Environmental Information (NCEI) as its Integrated Surface Dataset (ISD), see (Smith et al, 2011). The HadISD data have been quality controlled to remove random errors, with some station merging and duplicate removal to create long-term records.

Initially, datasets of 56 stations were provided, many of them with large gaps. After removing data before the largest gap in each of them, 55 stations were selected (the 56th being too short after truncation). We have removed large gaps, i.e., intervals with absent data - these contain no statistics and it is not clear how to use them for EWS analysis. Filling large gaps with interpolated data would affect autocorrelations and thus introduce undesirable bias in early-warning indicators, which would distort searches for instrumental changes in the data.

One of the motivations for this study was that in many cases of long-term climatological observations, metadata might be missing or difficult to obtain (some documentation may still be not digitalised).

While there were station-level measurements of pressure (variable 'stnlp'), these records were often short or patchy and the data quality was quite poor. Instead of these station pressure measurements, we used a climatological surface pressure from the nearest gridbox of the ECMWF ERA5 (Reanalysis v5) product.

Using the HadISD dewpoint & temperature variables and pressure variables from the ECMWF ERA5 for the considered area (Hersbach et al, 2020) (the reanalysis pressure variable, after necessary temporal interpolation, being an acceptable approximation for the purposes of our data processing), we obtained the relative humidity using equations (3).

Dewpoint and relative humidity datasets were pre-processed as follows:

- Where large gaps were observed, the initial part of the data that included the gap was removed

- Global trend and seasonality were removed using SSA (singular spectrum analysis) (Broomhead and King, 1986) from both dewpoint and relative humidity variables. SSA decomposes the input time series into a sum of components (low-frequency trend, seasonal modulations, and detrended noise). The method is based on the singular value decomposition of a special matrix constructed from the time series. The advantage of the method is that neither a parametric model nor stationarity conditions are to be assumed for the time series. This makes SSA a versatile tool of time series analysis. Detrending using SSA can affect the level of autocorrelations but not the trend in the indicator time series. EWS is estimated as monotonous positive trend, and this trend is present in both indicators of the raw and detrended data.

- Detrended fluctuations of dewpoint and relative humidity were analysed using lag-1 ACF for EWS for the purpose of detecting instrumental changes. The strength of the EWS trend was quantified using the Mann-Kendall coefficient for assessment of monotonous trend (ideally increasing trend gives Kendall value 1, ideally decreasing -1). For the considered 55 stations, averaged Kendall values of dewpoint detrended fluctuations are $0.30 \pm 0.54$, whereas for relative humidity detrended fluctuations, the Kendall values are $0.68 \pm 0.44$ (mean and standard error). This is summarised in Fig. 1.

Pre-processing of data may influence autocorrelations: when a smoothing filter is applied, correlations increase, which may lead to very high level of fluctuations of autocorrelation function close to critical value 1. Such EWS values are usually not informative, and detection of sensor changes would be unfeasible. However, in our paper we apply detrending, which acts in the opposite way to smoothening: detrending removes the low-frequency trend, and autocorrelations diminish.

Despite these modifications, the main object of interest is the monotonic trend in the EWS indicator, which denotes the change in the dynamics of fluctuations. The absolute values of the EWS indicator are less important in this context.

Significant negative trends in both variables were observed in datasets for the locations of Leeming, Fylingdales, Lyneham, Jersey. Such effects in early-warning indicators may denote stabilisation (opposite to a critical transition), which may be, for example, due to urban effects on the instruments, such as loss of shrubs and building up of green areas.

Autocorrelations were analysed using a short single window. Conventionally, a much larger window size is used for tipping point analysis, as the phenomenon of interest in geophysics is usually of longer scale (decades), often related to climate change. In this case, we need to consider short-term events (replacement of an instrument or its drift/calibration, station moves, local environment changes), and a short window with a smaller subset of data for averaging provide higher sensitivity for the purpose of our analysis. The size of the sliding window should be selected in such a way that high-frequency fluctuations (which produce maximal lag-1 autocorrelations) would be smoothed out, whereas intrinsic EWS would be detected (usually increasing in the

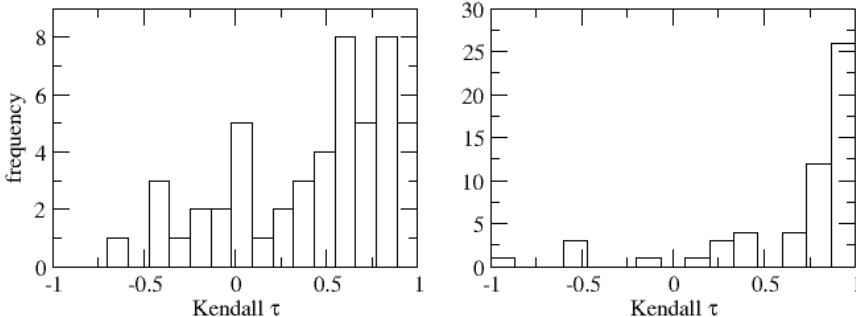

**Figure 1.** Histograms of Mann-Kendall coefficients of monotonic trends in early-warning indicators for dewpoint (left panel) and relative humidity (right panel). The window size for calculation of indicators was 10% of time series length (as the records had different lengths). High positive values of the coefficient denote significant increasing trend indicating EWS.

interval [0.7,1] — see the examples in the Appendix). The choice of window can be explored and furthermore automated using statistical techniques (such as scaling analysis with long-term exponents), because each geophysical variable has signature scaling behaviour, and therefore need to be investigated individually. In this case study, we note that we have selected a suitably small window size, which produced detectable EWS for the purposes of our analysis. In general, the smaller the window size, the better for studying sensor phenomena, as we want to identify nonstationarities, possibly caused by instrumental artefacts and sometimes reversed short-term transitions (for example, transient extreme events). In principle, it is possible to study the same time series with different windows of early-warning indicators, for different purposes: shortest one to identify instrument replacement, and longer one for studying environmental changes, such as encroachment of town or forest. EWS indicators have sensitivity for detection of short-term trends. However, such an indicator can still contain signatures of longer-term trends, although with noisy patterns. The choice of the window requires balancing of the sampling rate of time series and the dynamics of the phenomenon of interest. For example, for studying climate change effects, which manifest at the scale of several decades, it is necessary to consider an indicator with sufficiently large sliding window; for studying rapid changes in the time series, which may be due to instrument change or sensor deterioration, much smaller window size is necessary. Indicators with large window size usually have smoother patterns due to aggregation and average of large subsets of data.

Spatial distribution of the detected long-term changes in the EWS indicators across the UK for relative humidity is mapped according to the Mann-Kendall coefficient for detection of monotonous trends, which is illustrated in Fig. 2. This plot shows where the long-term changes in humidity are more pronounced (large blue dots), and this can be further analysed in terms of local micro-climates and city/country distributions.

We consider the Bingley station (Midlands countryside) as a representative example for detailed analysis. It is known that there was a period of instrumental changes in environmental sensors. For example, in the 1980s-1990s there were replacements of the earlier analogue instruments with modern digital, which can be detected using ACF-based indicators of the tipping point analysis. Figure 3 illustrates at a glance such a clear change. It is interesting that the probability density functions of dew point

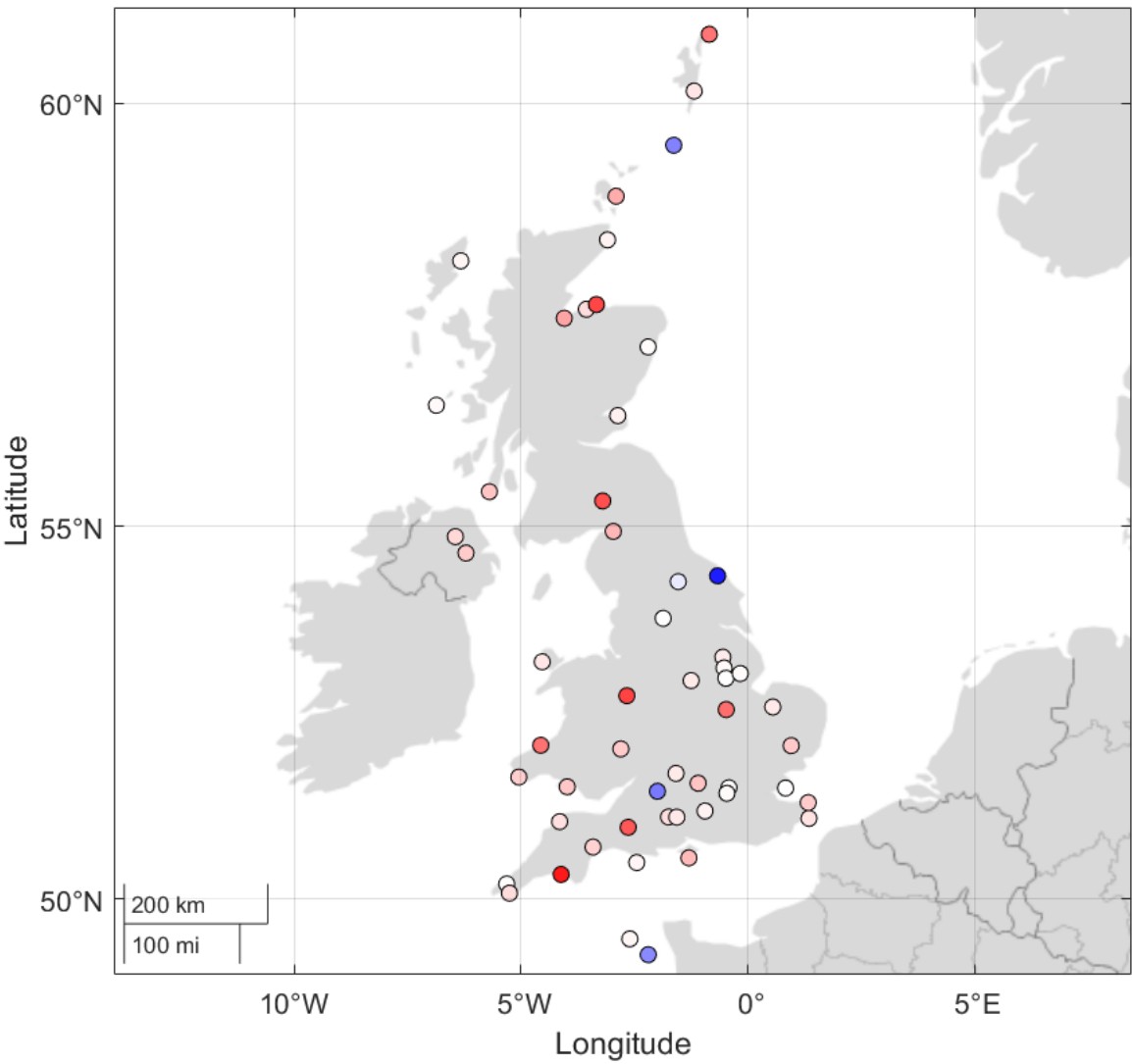

**Figure 2.** Spatial mapping of linear trends in EWS indicators for the detrended fluctuations of relative humidity in the UK. Trends were fitted over the whole range of each indicator. The colour denotes direction of trend (red — positive, blue — negative). The brighter colour of dot is defined by the value of the lead coefficient: darker red for higher positive value, darker blue for lower negative value.

and RH have different shapes (skewness), probably due to the nonlinearity of the equations linking them, as can be seen in
Fig. 4. Autocorrelation functions differ, too, which reflects memory in the data (i.e., internal dependencies between the points of time series), as can be seen in Fig. 5.

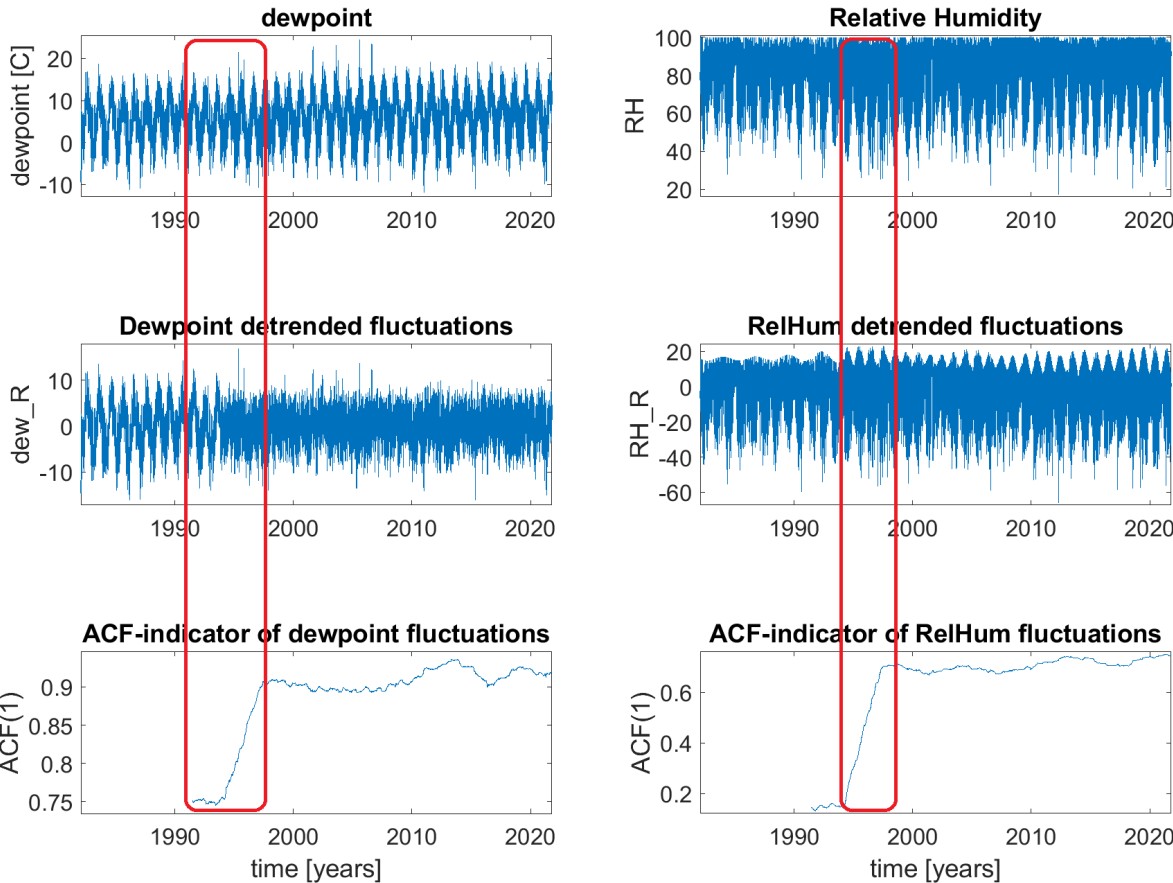

**Figure 3.** Bingley dew point and relative humidity (upper panels), their detrended fluctuations (middle panels) and EWS indicators calculated with 10% sliding windows. The red boxes denote the intervals of transitions: in the input time series the changes are not visible, whereas in the detrended series one can notice the change of pattern, which is then clearly detected by the EWS indicator — the likely instrumental change in 1995.

To perform automatic identification of instrumental changes, one can use probabilistic estimation of change points in the EWS indicators. By applying the Matlab package "Bayesian Changepoint Detection & Time Series Decomposition" (Zhao et al, 2019), we identify change points in EWS indicators, specifically those of statistical significance. As can be seen in the following plots, the changes in the indicators that are statistically significant can be identified in the probabilities of the changes based on the Bayesian ensembles with Monte Carlo simulations (Zhao et al, 2019). The considered variable was detrended relative humidity, for which the ACF-indicator was calculated, which then was analysed using the Bayesian model ensemble. The changes with statistically significant peaks (probability >0.8) are narrow, and the timing of change events can be identified with small uncertainty of several weeks, if necessary.

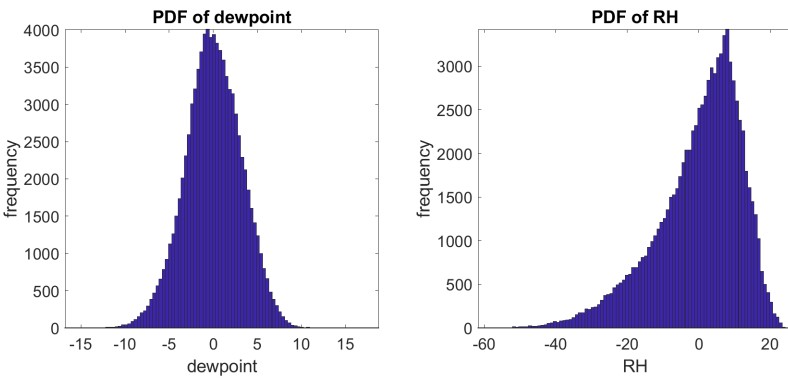

**Figure 4.** Histograms of the Bingley dew point and relative humidity.

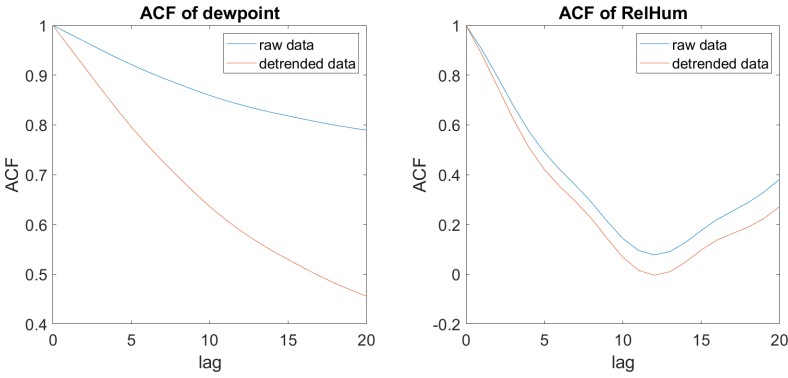

**Figure 5.** Autocorrelation functions of the Bingley dew point and relative humidity, with and without detrending.

Because the series vary in length, while we are using the window length of 10% of the series length, the sliding window aggregates data of different length in each case. They are comparable but vary from series to series. Furthermore, there is an uncertainty due to aggregation of data within a selected window. This uncertainty can be reduced by using smaller and smaller window, if there is a task of precise timing of the change. In this paper, we aim to demonstrate that such detection is possible in principle, and in further work we can consider the problem of timing with reduced uncertainties.

In the case of Bingley, there are two statistically significant changes, in 1994 and in 1998, which are indicated in the lower panel with probabilities of changes, see Fig. 6. The red dots in the upper panels denote the recorded changes of operations (this data was provided by Met Office for verification), for which the major changes from manual to automatic were detected by the abrupt increases in indicators. The introduced method of instrumental recording changed the patterns of autocorrelations, and this is detectable by the EWS indicators. In the later period, short intermittencies did not affect the indicators as much as the

major change of the technology in the 1990s, yet many of them can be seen in the probabilistic detections in the bottom panel of Fig. 6.

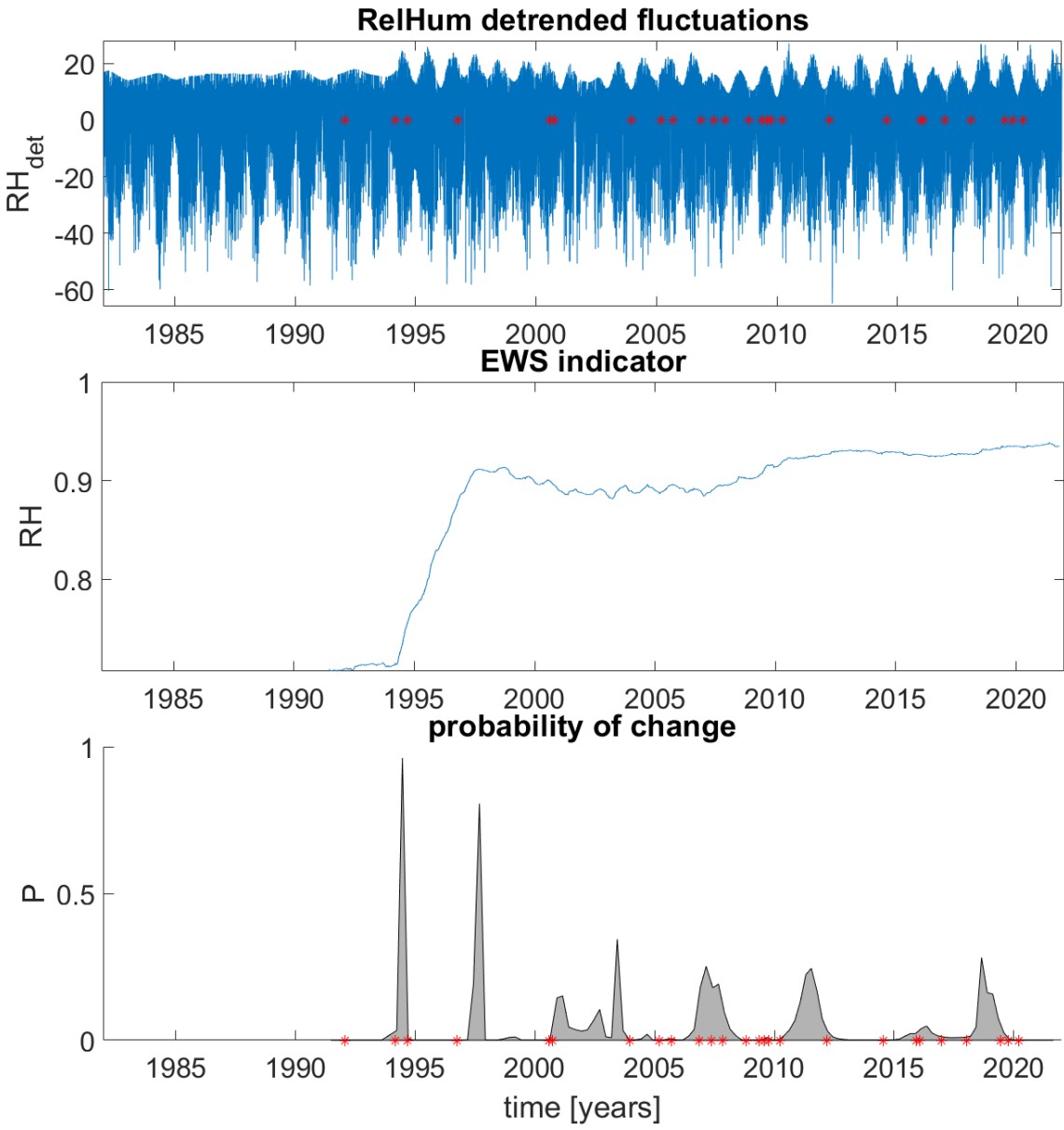

**Figure 6.** Bingley detrended (det) fluctuations of relative humidity, its ACF-indicator with a small window, which is suitable for detection of instrumental changes with change of fluctuation patterns (middle panel), and Bayesian analysis of the indicator time series, which denotes probabilities of changes (lower panel). Red dots in the lower and upper panels denote the recorded changes of operations.

In further discussions of the known instrument changes, it was mentioned that in Bingley, in 1994 and 1996, there were changes from manual to automatic, as well as a change of the local airfield, which is an excellent confirmation of the detected abrupt transition. Moreover, the smaller detections in the years 2007, 2011, 2018 were reported to be related to routine re-
250 placements of sensors at Bingley. This demonstrates the capability of these techniques in sensitive detection of instrumental change.

For comparison, we plot equivalent results for stations Camborne (Fig. 7) and Carlisle (Fig. 8). We note that both of these analyses show strong detections of changes in the 1990s.

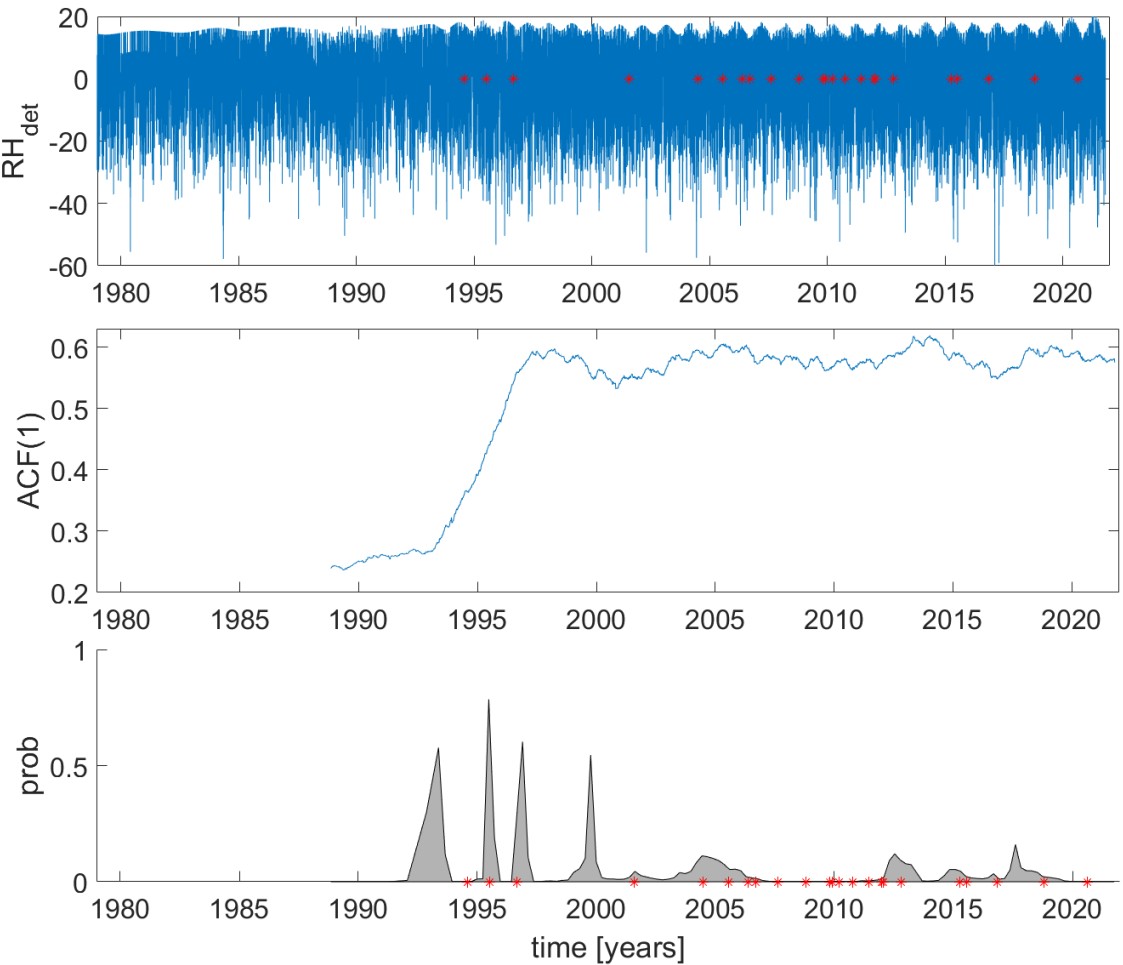

**Figure 7.** Camborne detrended (det) fluctuations of relative humidity, its ACF-indicator with a small window, which is suitable for detection of instrumental changes with change of fluctuation patterns (middle panel), and Bayesian analysis of the indicator time series, which denotes probabilities of changes (lower panel). Red dots in the lower and upper panels denote the recorded changes of operations.

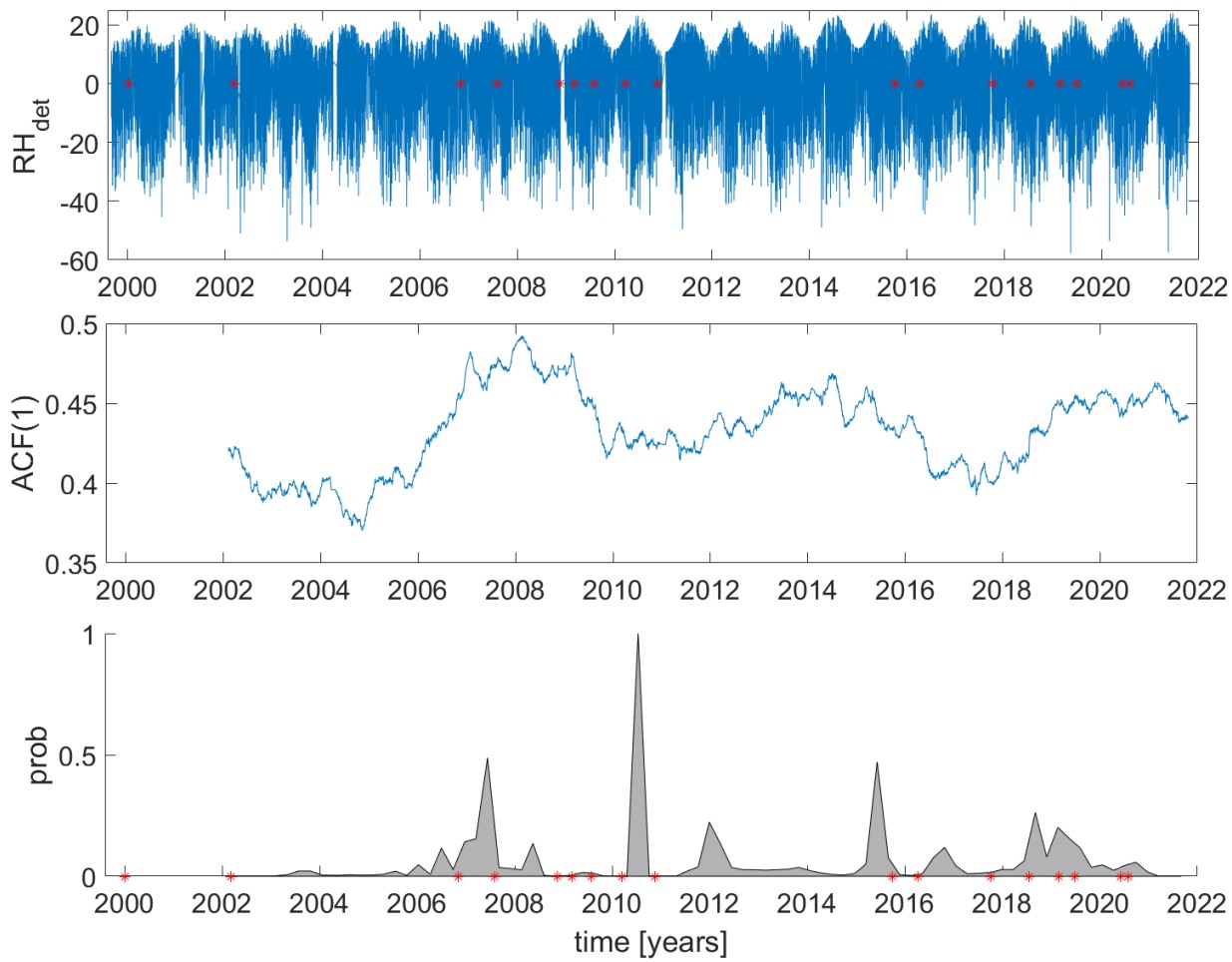

**Figure 8.** Carlisle detrended (det) fluctuations of relative humidity, its ACF-indicator with a small window, which is suitable for detection of instrumental changes with change of fluctuation patterns (middle panel), and Bayesian analysis of the indicator time series, which denotes probabilities of changes (lower panel). Red dots in the lower and upper panels denote the recorded changes of operations.

The results of the analysis of the changes in the indicators for relative humidity for the large set of 55 observing stations
are summarised in the supplementary Table in the Appendix. Abrupt changes in EWS indicators are likely to be instrumental, whereas gradual ones are likely to be climatic (either natural, such as long-term climate change, or anthropogenic, such as urbanisation and change of local environment). The reported detections were later confirmed by available records of instrumental changes for some of the stations.

To complement the analysis based on autocorrelations, we also demonstrate early warning signals based on variance (see
figures B1-B22 in the Appendix). (Smith et al, 2023) previously discussed changes in measurement noise and demonstrated

that they may lead to anti-correlation between AR1 and variance indicators. This confirms the earlier observations by (Livina et al, 2012) that increasing correlations in the data may lead to increase of autocorrelations and simultaneous decrease of variance. In particular, this is due to the final-size effect of the observational sliding window used for calculation of each indicator. In our current analysis, the variance indicator provided noisier results with less clear changes than autocorrelations, and therefore the autocorrelation indicator seems more suitable for the purpose of instrumental change detection.

## 4    Conclusion

We have applied tipping point analysis in a novel way (compared with its conventional use) to study instrumental and sensor phenomena in a large UK dataset of relative humidity. We carefully preprocessed the data to ensure that no underlying trends would affect the results, and as a consequence we have been able to reveal signatures of sensor events and phenomena.

In many of the stations, when applying a small sliding window, various transitions were observed. In particular, in the mid-1980s and mid-1990s (see figures in the main text and in the Appendix), the tipping point analysis identified several rapid transitions, which are likely to have been caused by instrumental artefacts, based on the verification from the station records. In other cases, transitions are gradual, and often reversed later. Such reversal changes may be related to local stabilisation of environmental conditions.

We did not attempt to distinguish different types of sensor issues. Rather, we wanted to demonstrate that general detection is possible, which can then guide further investigation of such issues.

In this paper, we analysed dewpoint and relative humidity In some cases, relative humidity is measured directly, but in the dataset we use it has been calculated using converion to dew point temperature, with several stages of processing. We identified short-term instrumental effects in the data using early-warning indicators of the tipping point analysis. These techniques are useful in identifying instrumental changes in those cases, when documentation and historical metadata may be missing. This demonstration of the application of the early-warning signal techniques is new and supplementary to their conventional use of tipping point analysis in climatology and geophysics.

We have demonstrated that autocorrelations are both sensitive and robust in detection of known sensor changes. While not all metadata of measurement circumstances may be available, autocorrelation indicators provide a tool for scanning datasets for such changes, whose rapid development may help distinghuish them from signals generated by slower processes, such as climate change. This dynamics should be taken into account in development of predictive techniques based on EWS signatures.

Our approach has the advantage of detecting changes that are not necessarily related to standard deviation, as many such changes are related to autocorrelations only. The method is computationally light and does not require extensive model training with tuning of multiple parameters.

The observed effects are not climatic tippings, but rather an example of short-term critical transitions caused by instrumental modifications and/or sensor artefacts. Such changes are important to identify, and our approach allows their automation and, if necessary, in real-time. This means that EWS techniques could be used for condition monitoring of environmental sensor networks. This is a promising novel application of the tipping point analysis in a new domain.

**Acknowledgement**

This work was funded by the UK Government's Department for Science, Innovation & Technology (DSIT) through the UK's National Measurement System programmes. Kate Willett was supported by the Met Office Hadley Centre Climate Programme funded by DSIT. We would like to thank the reviewers for their comments that helped improve the manuscript.

## 4.1 Code/Data availability

The code is based on the autocorrelation function that is available in any programming language. The data is property of Met Office UK.

## 4.2 Author contribution

The authors contributed equally to manuscript development.

## 4.3 Competing interests

The authors declare that they have no conflict of interest.

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

# Appendix

We summarise the analysis in the Table and provide further illustrations from several other stations.

| N | Station | Year of detected change | Likely cause |
|---|---------|------------------------|--------------|
| 1 | Aberporth | 2011 | climatic |
| 2 | Aldergrove | 1984, 1989 | instrumental |
| 3 | Baltasound | - | - |
| 4 | Benson | 2004, 2006, 2019 | climatic |
| 5 | Bingley | 1994, 1997 | instrumental |
| 6 | Boscombe Down | 2008, 2013 | climatic |
| 7 | Brize Norton | 1982, 1989 | instrumental |
| 8 | Camborne | 1995 | instrumental |
| 9 | Carlisle | 2011 | climatic |
| 10 | Chivenor | 1995, 2004, 2006, 2013 | climatic |
| 11 | Coningsby | 1983 | instrumental |
| 12 | Cranwell | 1996 | instrumental |
| 13 | Culdrose | - | - |
| 14 | Dyce | 1982 | instrumental |
| 15 | Eskdalemuir | 1984 | instrumental |
| 16 | Exeter | - | - |
| 17 | Fair Isle | - | - |
| 18 | Fylingdales | - | - |
| 19 | Guernsey | - | - |
| 20 | Heathrow | 1982 | instrumental |
| 21 | Hereford-Credenhill | 2005 | climatic |
| 22 | Inverness | 1995, 2000 | climatic |
| 23 | Isle of Portland | 1998, 1999 | instrumental |
| 24 | Jersey | 2001 | climatic |
| 25 | Kinloss | 2017, 2018 | climatic |
| 26 | Kirkwall | - | - |
| 27 | Langdon Bay | 1989 | instrumental |
| 28 | Leeming | - | - |
| 29 | Lerwick | - | - |
| 30 | Leuchars | 1983 | instrumental |
| 31 | Linton on Ouse | - | - |
| 32 | Lossiemouth | 2006, 2008 | climatic |
| 33 | Lyneham | - | - |
| 34 | Machrihanish | 2014, 2016, 2017 | climatic |
| 35 | Manston | 1982 | instrumental |
| 36 | Marham | - | - |
| 37 | Middle Wallop | 1988 | instrumental |
| 38 | Milford Haven Conservancy Boa | 2007 | climatic |
| 39 | Mumbles Head | 2011, 2015 | climatic |
| 40 | Northolt | 1984 | instrumental |
| 41 | Nottingham-Watnall | - | - |
| 42 | Odiham | 1988, 1995, 2007 | instrumental |
| 43 | Plymouth-Mountbatten | 2014 | climatic |
| 44 | Portglenone | 2007 | climatic |
| 45 | Scampton | 2018 | climatic |
| 46 | Shawbury | - | - |
| 47 | Shoeburyness-Landwick | 1985 | instrumental |
| 48 | Tiree | - | - |
| 49 | Valley | 1980 | instrumental |
| 50 | Waddington | 1982, 2004, 2008 | instrumental |
| 51 | Wattisham | 1989 | instrumental |
| 52 | Wick | - | - |
| 53 | Wight - St. Catherines Point | 2001 | instrumental |
| 54 | Wittering | 2020, 2021 | climatic |
| 55 | Yeovilton | 1995, 1998, 2010, 2016 | climatic |

400

**Table A1.** Detection of changes in the UK relative humidity records based on the detections probability above 0.8 using the Bayesian technique.

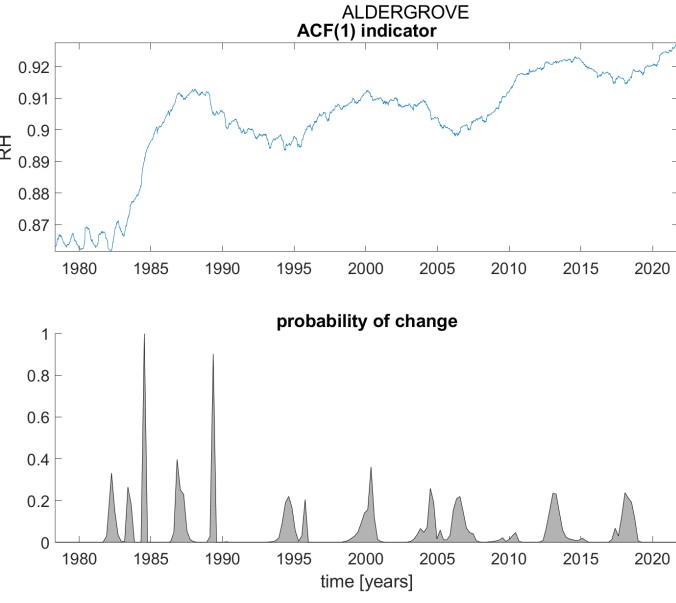

**Figure A1.** ACF(1) indicator (upper panel) and probability of detection of changes in the ACF(1) indicator (lower panel) for station Aldergrove.

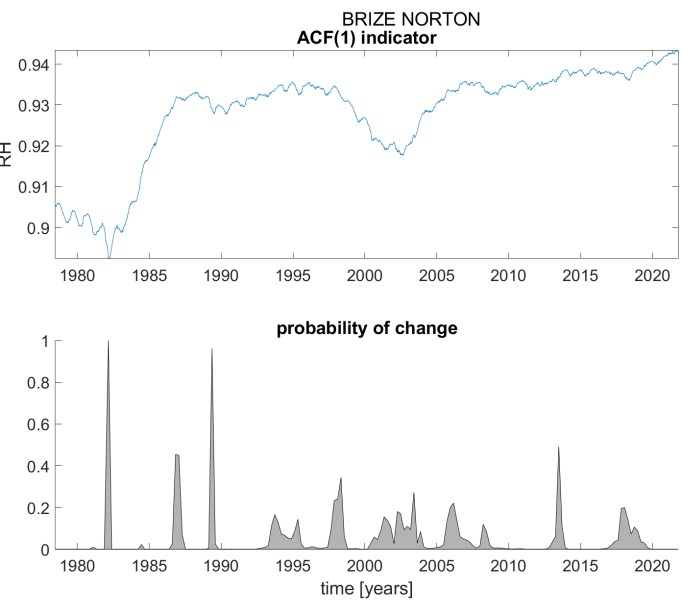

**Figure A2.** ACF(1) indicator (upper panel) and probability of detection of changes in the ACF(1) indicator (lower panel) for station Brize Norton.

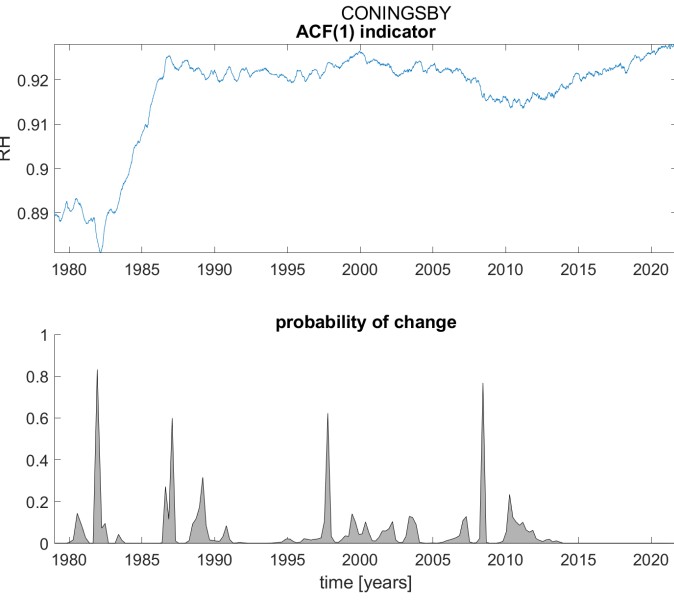

**Figure A3.** ACF(1) indicator (upper panel) and probability of detection of changes in the ACF(1) indicator (lower panel) for station Coningsby.

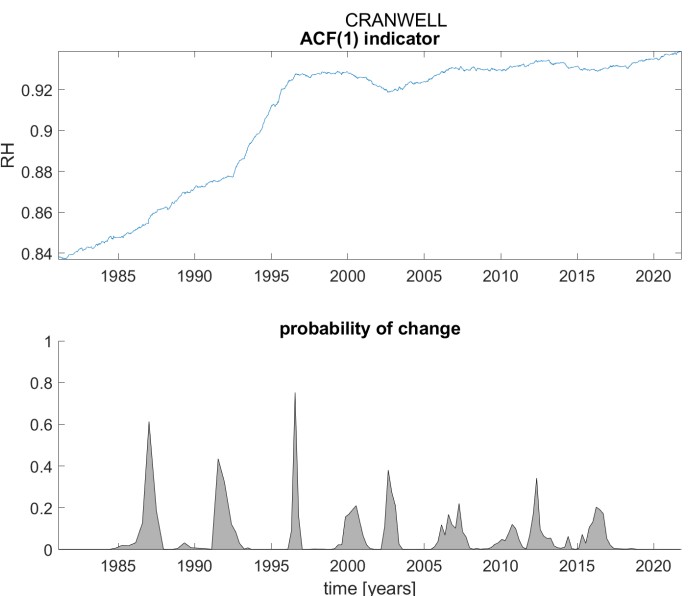

**Figure A4.** ACF(1) indicator (upper panel) and probability of detection of changes in the ACF(1) indicator (lower panel) for station Cranwell.

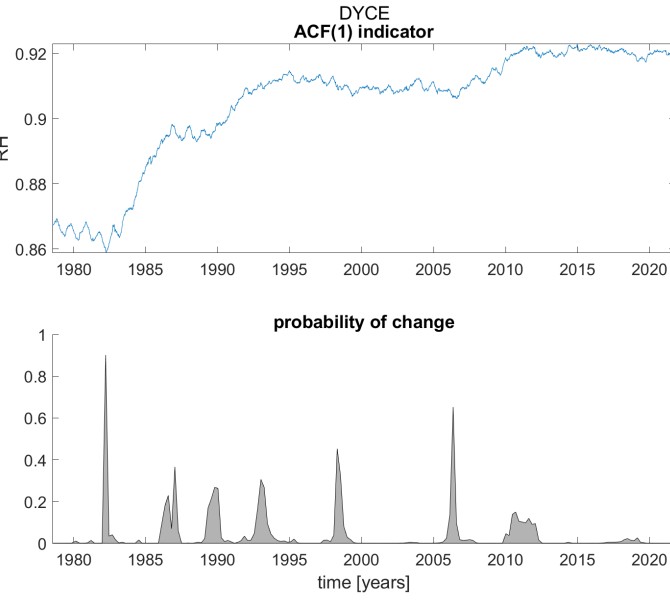

**Figure A5.** ACF(1) indicator (upper panel) and probability of detection of changes in the ACF(1) indicator (lower panel) for station Dyce.

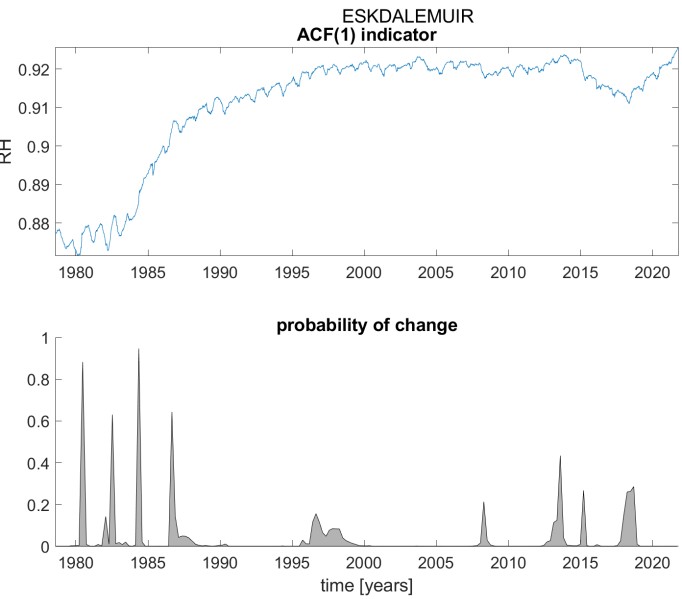

**Figure A6.** ACF(1) indicator (upper panel) and probability of detection of changes in the ACF(1) indicator (lower panel) for station Eskdale-muir.

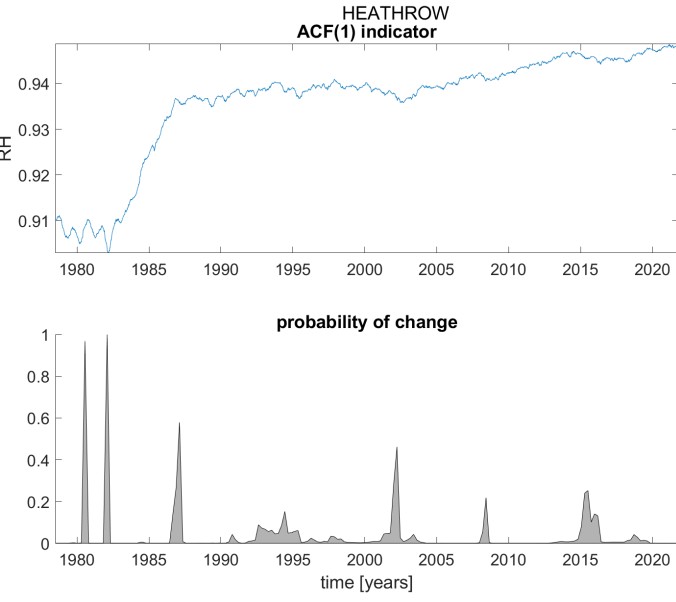

**Figure A7.** ACF(1) indicator (upper panel) and probability of detection of changes in the ACF(1) indicator (lower panel) for station Heathrow.

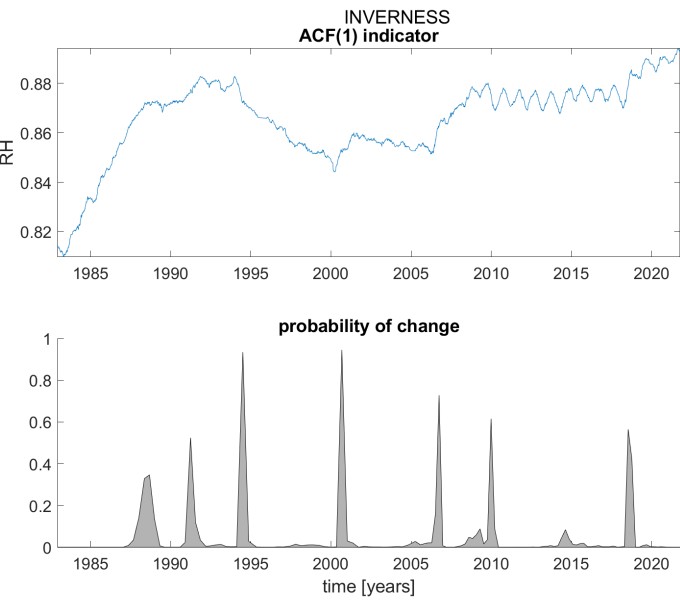

**Figure A8.** ACF(1) indicator (upper panel) and probability of detection of changes in the ACF(1) indicator (lower panel) for station Inverness.

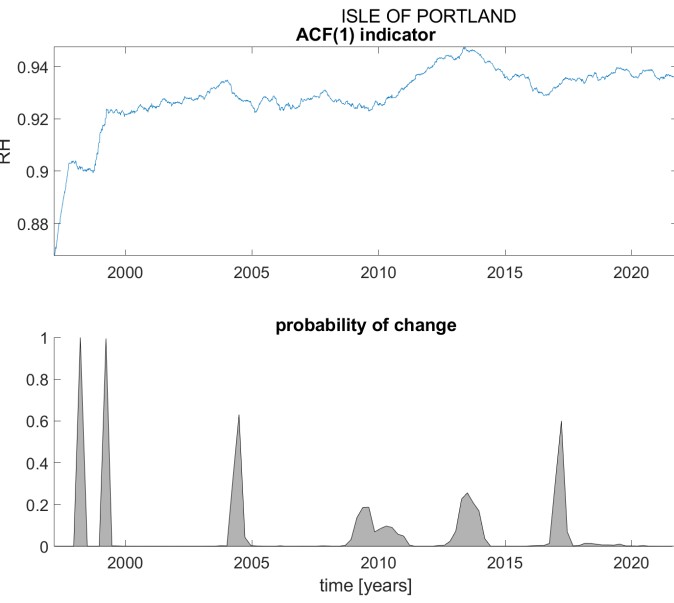

**Figure A9.** ACF(1) indicator (upper panel) and probability of detection of changes in the ACF(1) indicator (lower panel) for station Isle of Portland.

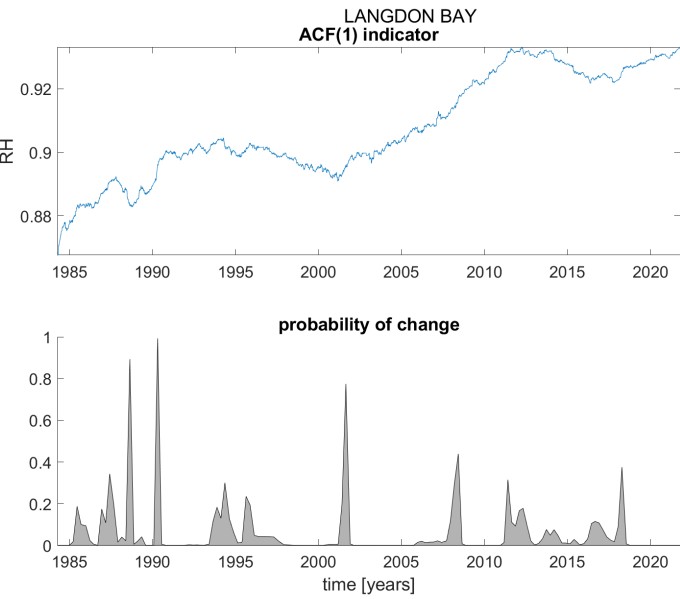

**Figure A10.** ACF(1) indicator (upper panel) and probability of detection of changes in the ACF(1) indicator (lower panel) for station Langdon Bay.

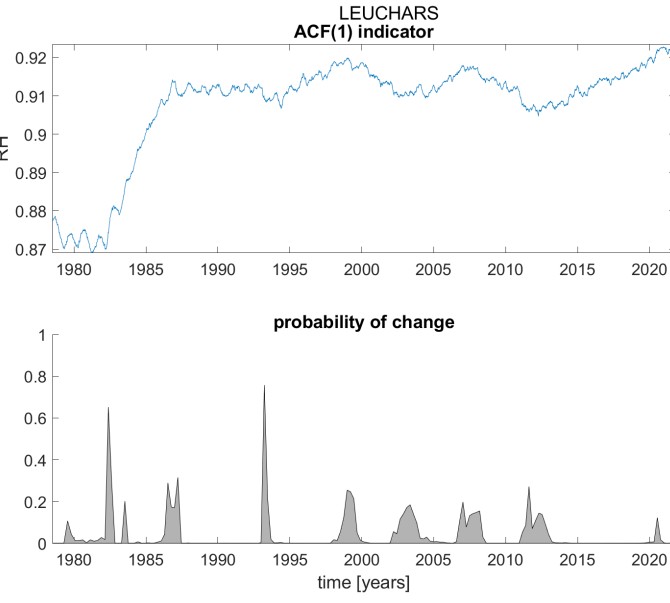

**Figure A11.** ACF(1) indicator (upper panel) and probability of detection of changes in the ACF(1) indicator (lower panel) for station Leuchars.

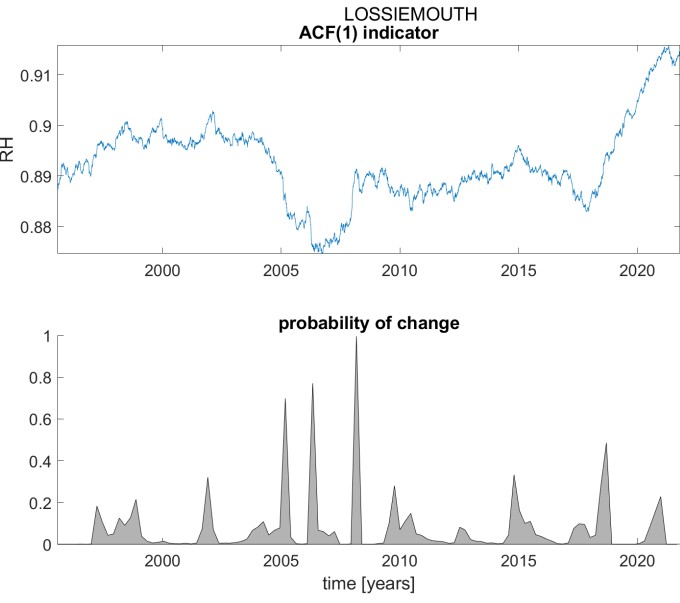

**Figure A12.** ACF(1) indicator (upper panel) and probability of detection of changes in the ACF(1) indicator (lower panel) for station Lossiemouth.

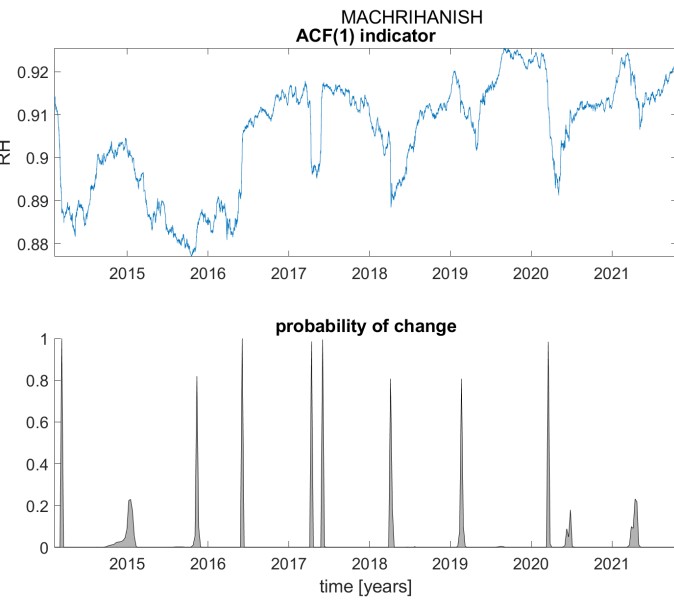

**Figure A13.** ACF(1) indicator (upper panel) and probability of detection of changes in the ACF(1) indicator (lower panel) for station Machrihanish.

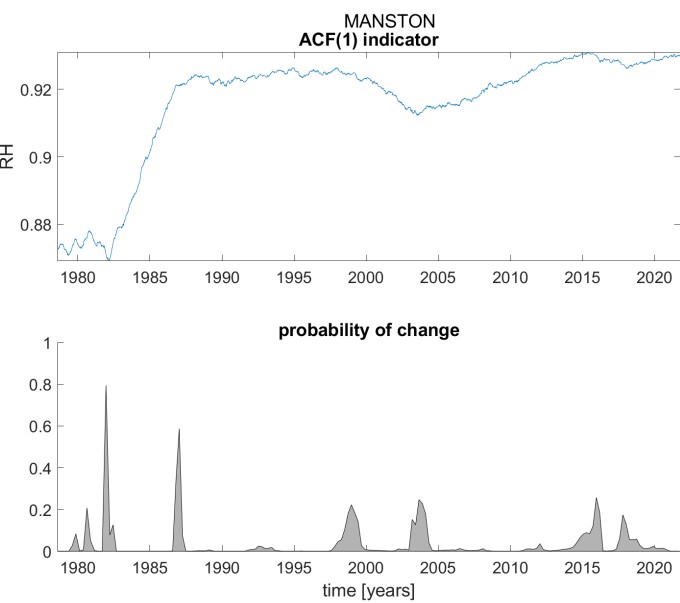

**Figure A14.** ACF(1) indicator (upper panel) and probability of detection of changes in the ACF(1) indicator (lower panel) for station Manston.

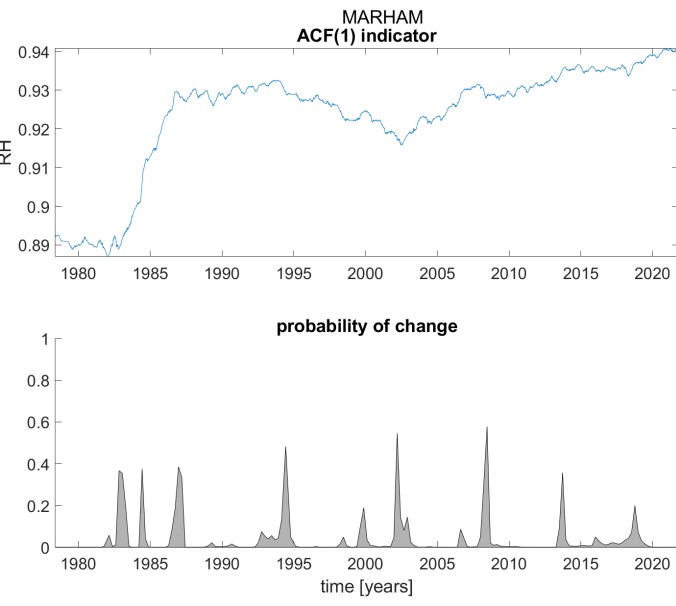

**Figure A15.** ACF(1) indicator (upper panel) and probability of detection of changes in the ACF(1) indicator (lower panel) for station Marham.

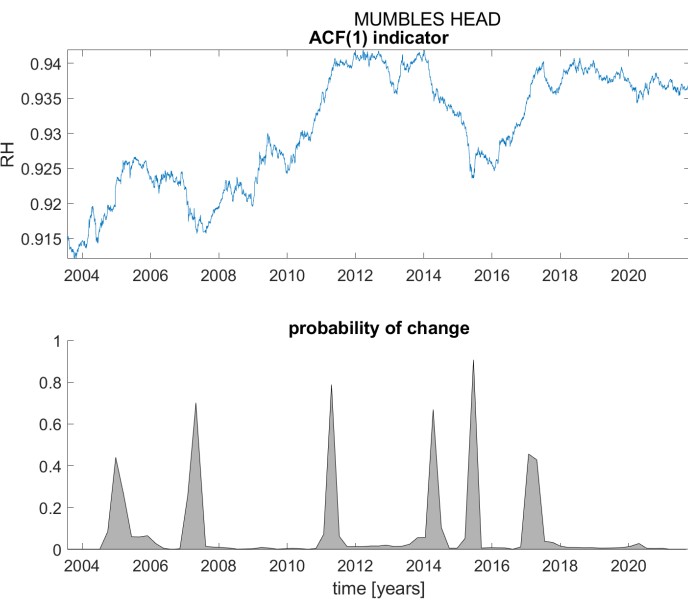

**Figure A16.** ACF(1) indicator (upper panel) and probability of detection of changes in the ACF(1) indicator (lower panel) for station Mumbles Head.

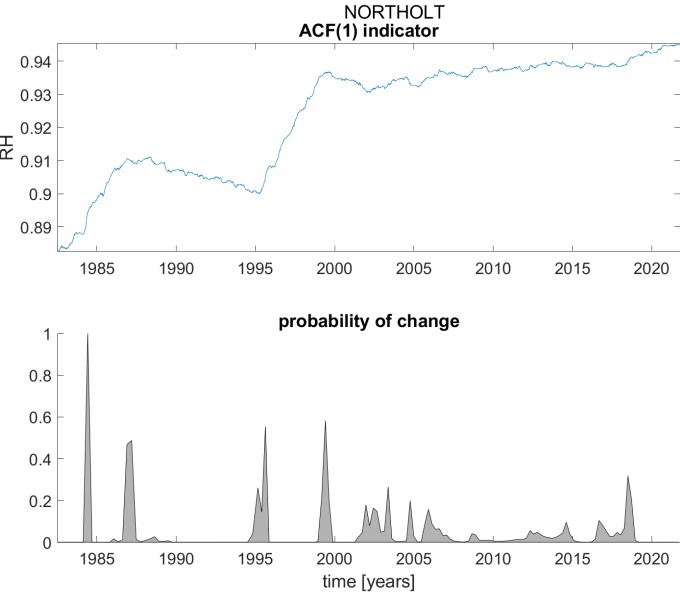

**Figure A17.** ACF(1) indicator (upper panel) and probability of detection of changes in the ACF(1) indicator (lower panel) for station Northolt.

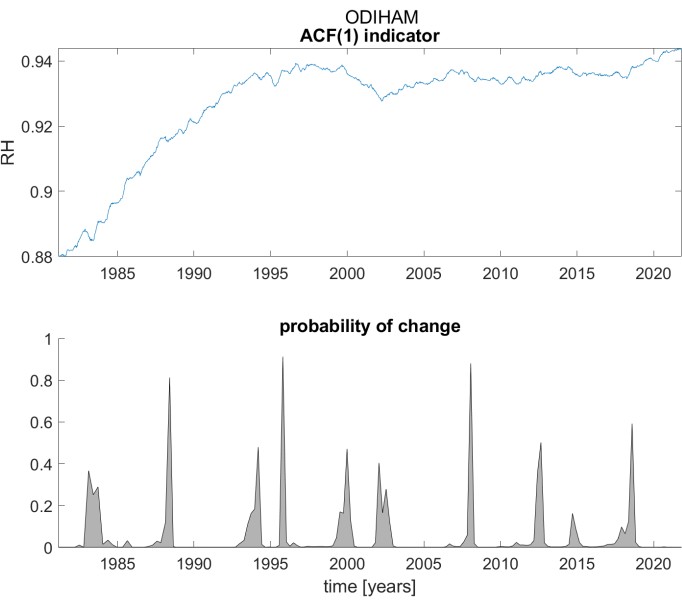

**Figure A18.** ACF(1) indicator (upper panel) and probability of detection of changes in the ACF(1) indicator (lower panel) for station Odiham.

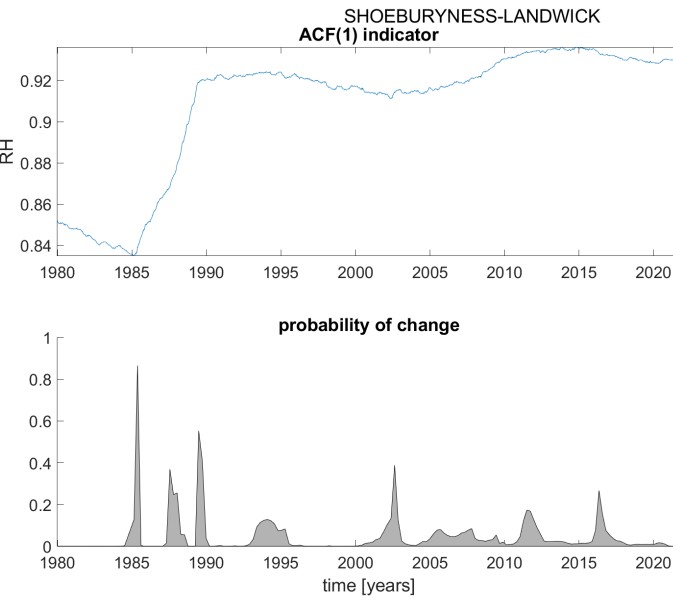

**Figure A19.** ACF(1) indicator (upper panel) and probability of detection of changes in the ACF(1) indicator (lower panel) for station Shoeburyness-Landwick.

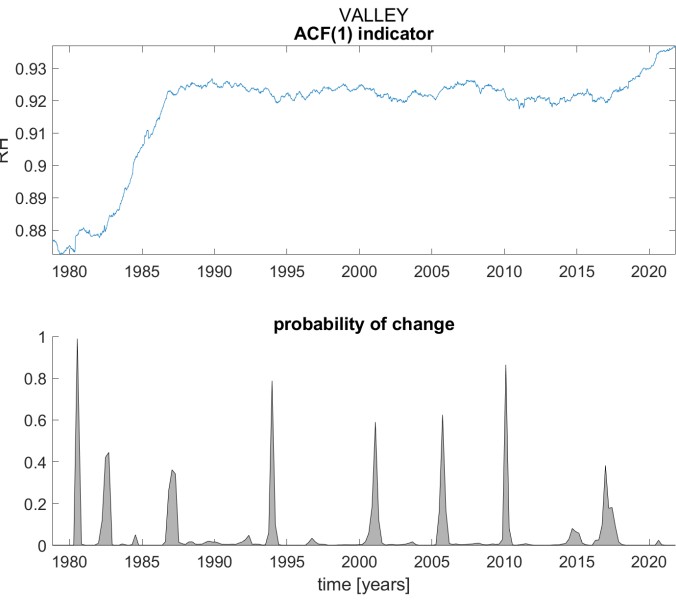

**Figure A20.** ACF(1) indicator (upper panel) and probability of detection of changes in the ACF(1) indicator (lower panel) for station Valley.

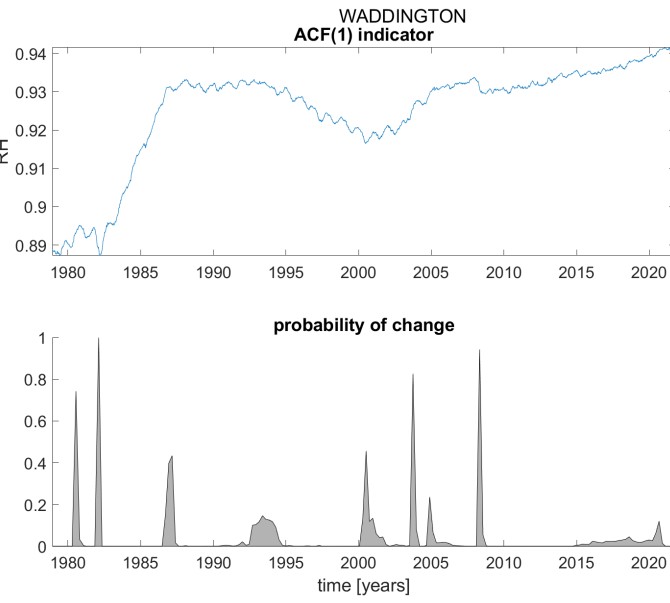

**Figure A21.** ACF(1) indicator (upper panel) and probability of detection of changes in the ACF(1) indicator (lower panel) for station Waddington.

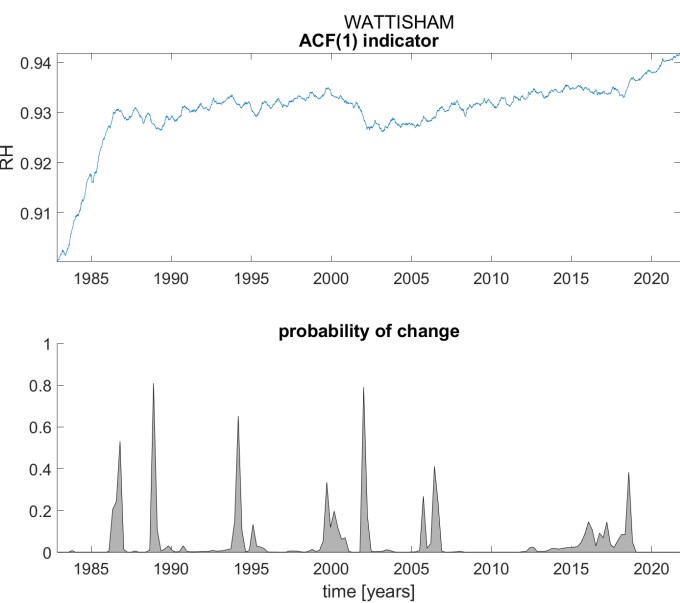

**Figure A22.** ACF(1) indicator (upper panel) and probability of detection of changes in the ACF(1) indicator (lower panel) for station Wattisham.

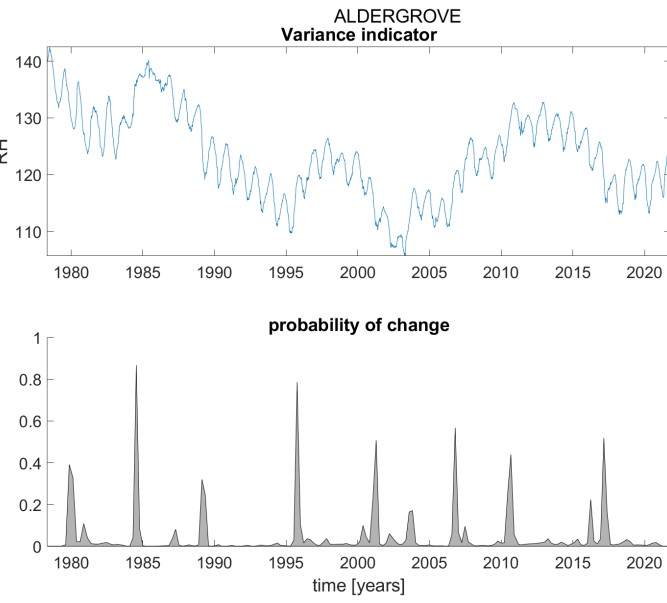

**Figure B1.** Variance indicator (upper panel) and probability of detection of changes in the variance indicator (lower panel) for station Aldergrove.

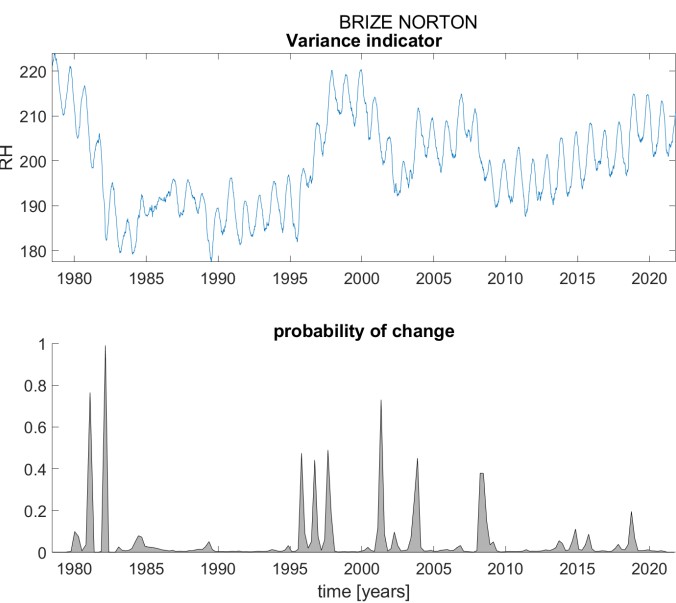

**Figure B2.** Variance indicator (upper panel) and probability of detection of changes in the variance indicator (lower panel) for station Brize Norton.

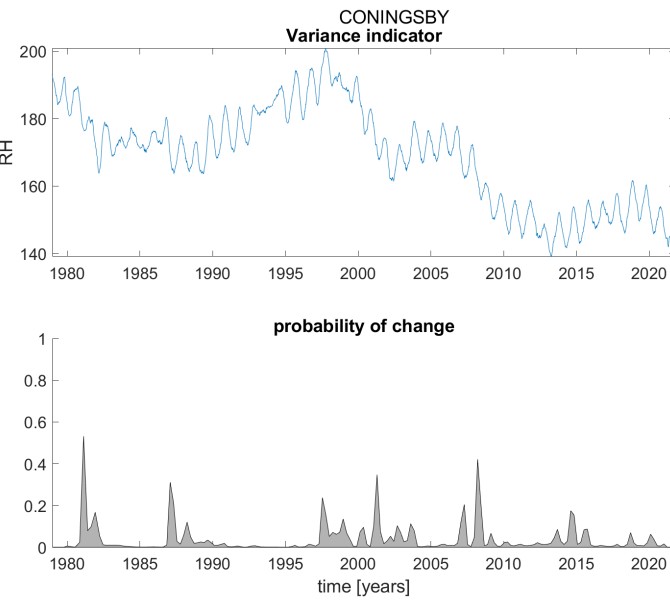

**Figure B3.** Variance indicator (upper panel) and probability of detection of changes in the variance indicator (lower panel) for station Coningsby.

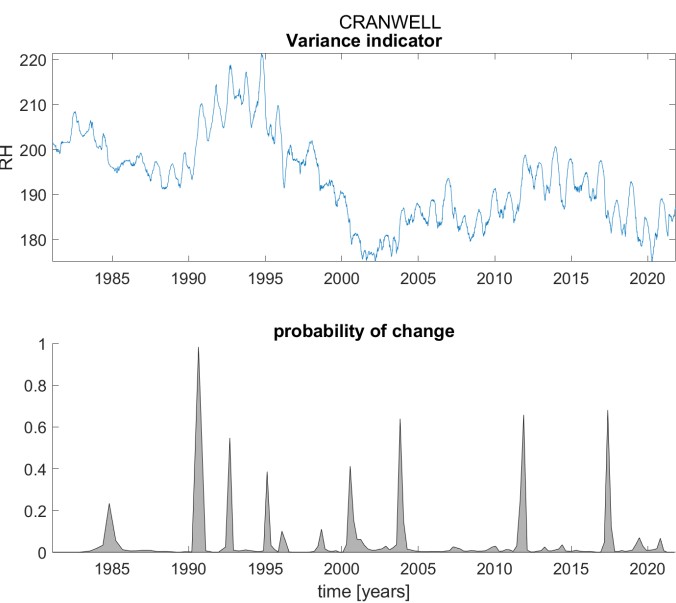

**Figure B4.** Variance indicator (upper panel) and probability of detection of changes in the variance indicator (lower panel) for station Cranwell.

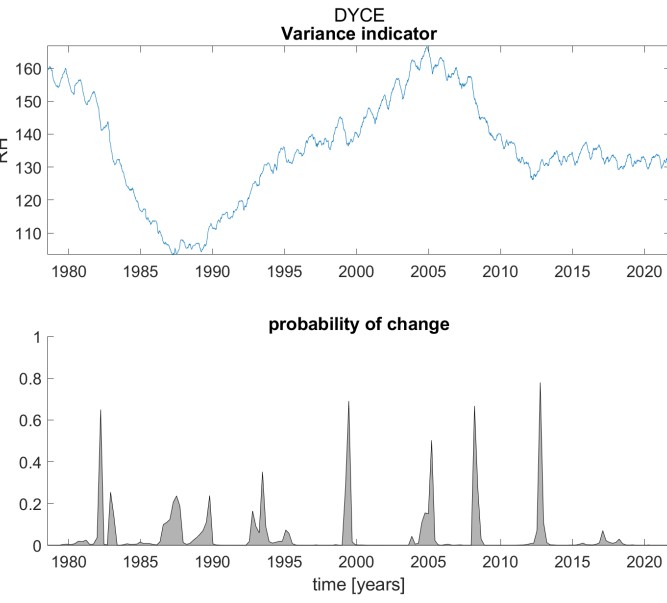

**Figure B5.** Variance indicator (upper panel) and probability of detection of changes in the variance indicator (lower panel) for station Dyce.

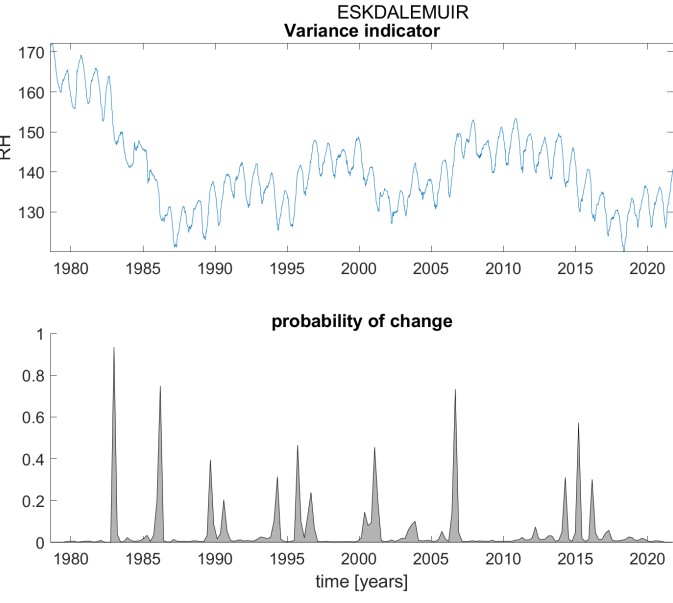

**Figure B6.** Variance indicator (upper panel) and probability of detection of changes in the variance indicator (lower panel) for station Eskdalemuir.

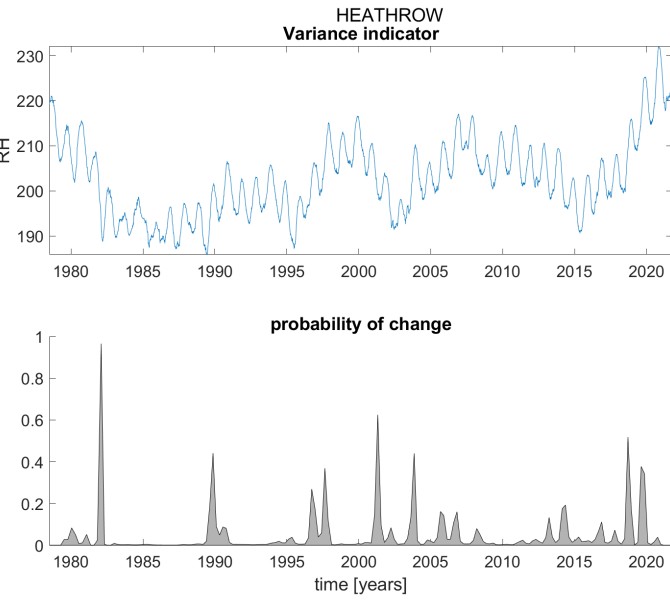

**Figure B7.** Variance indicator (upper panel) and probability of detection of changes in the variance indicator (lower panel) for station Heathrow.

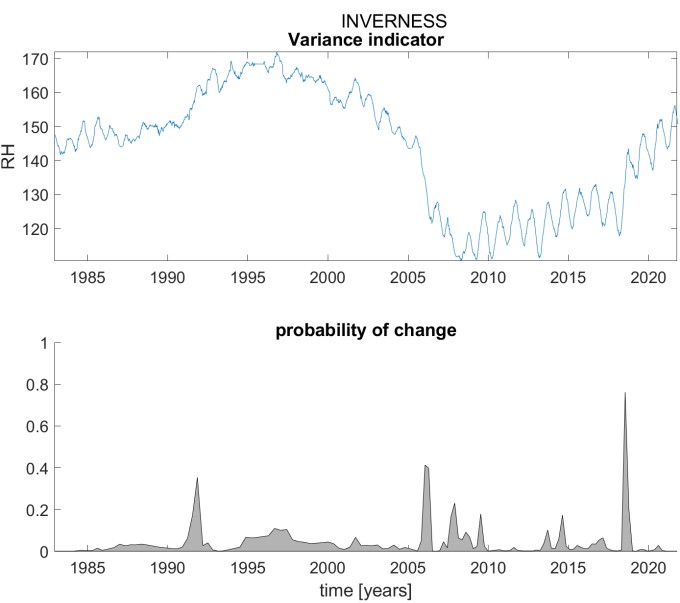

**Figure B8.** Variance indicator (upper panel) and probability of detection of changes in the variance indicator (lower panel) for station Inverness.

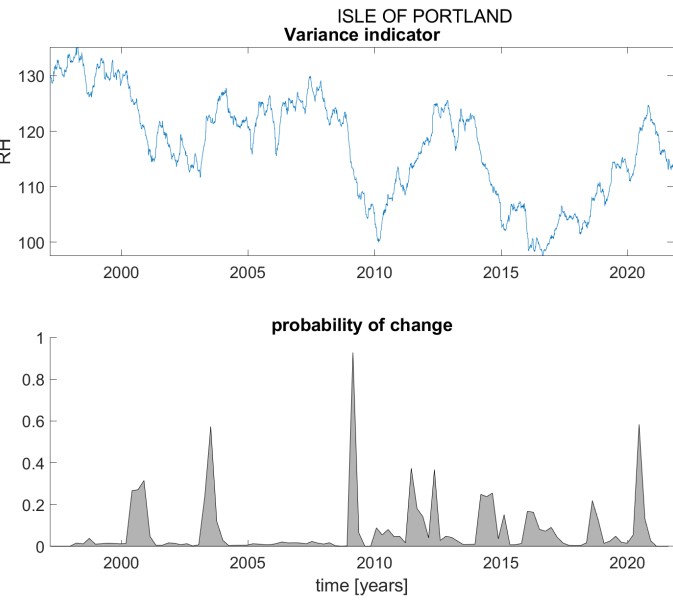

**Figure B9.** Variance indicator (upper panel) and probability of detection of changes in the variance indicator (lower panel) for station Isle of Portland.

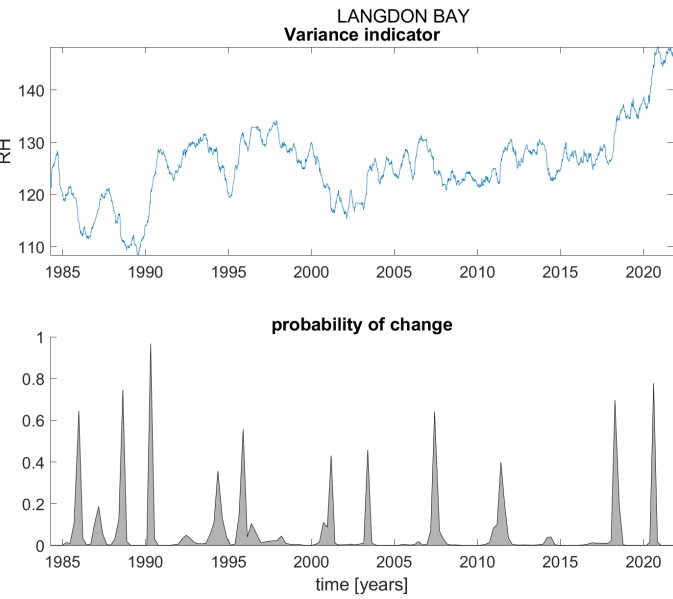

**Figure B10.** Variance indicator (upper panel) and probability of detection of changes in the variance indicator (lower panel) for station Langdon Bay.

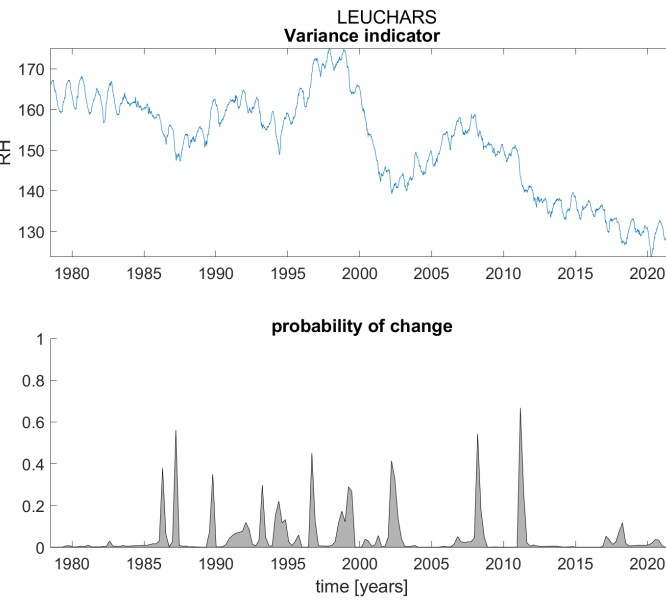

**Figure B11.** Variance indicator (upper panel) and probability of detection of changes in the variance indicator (lower panel) for station Leuchars.

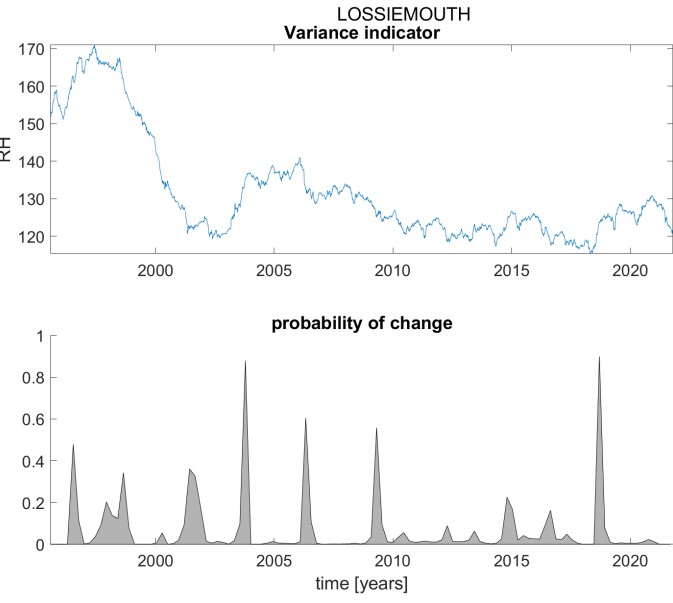

**Figure B12.** Variance indicator (upper panel) and probability of detection of changes in the variance indicator (lower panel) for station Lossiemouth.

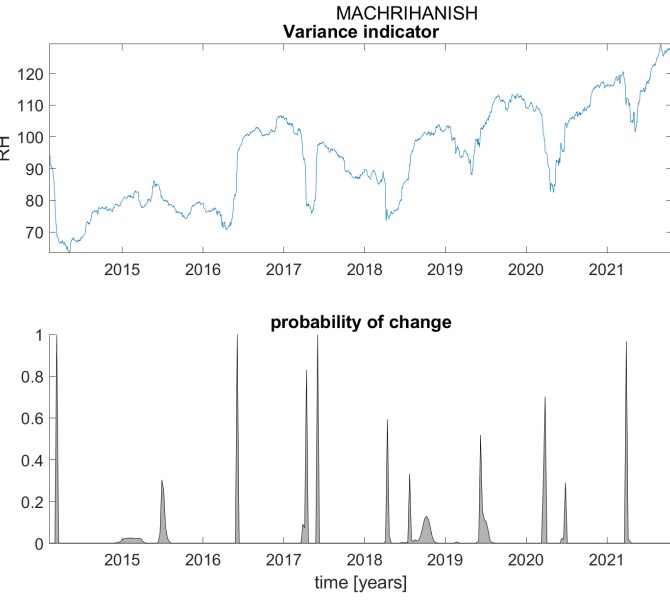

**Figure B13.** Variance indicator (upper panel) and probability of detection of changes in the variance indicator (lower panel) for station Machrihanish.

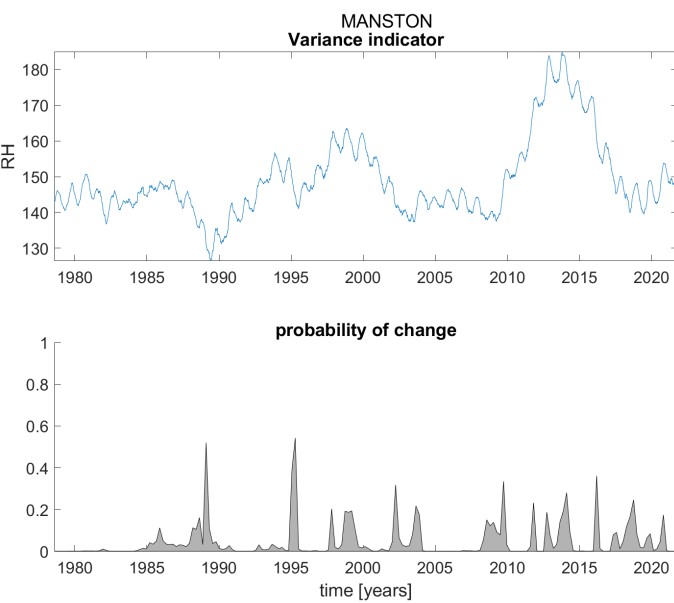

**Figure B14.** Variance indicator (upper panel) and probability of detection of changes in the variance indicator (lower panel) for station Manston.

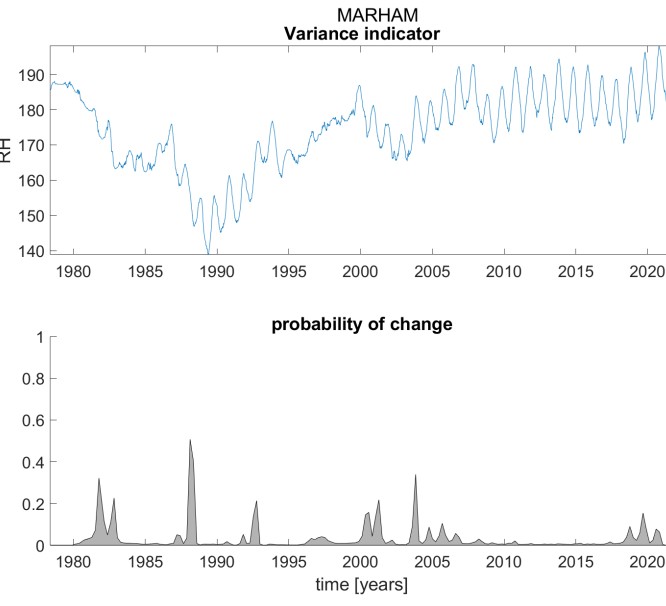

**Figure B15.** Variance indicator (upper panel) and probability of detection of changes in the variance indicator (lower panel) for station Marham.

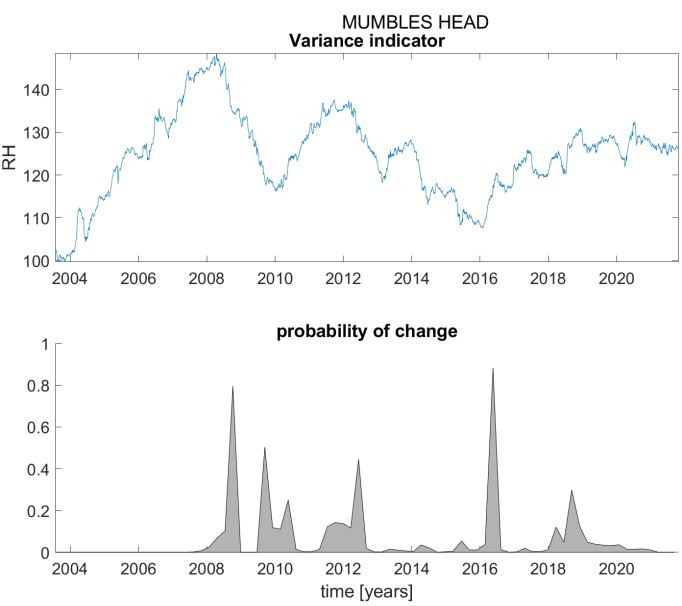

**Figure B16.** Variance indicator (upper panel) and probability of detection of changes in the variance indicator (lower panel) for station Mumbles Head.

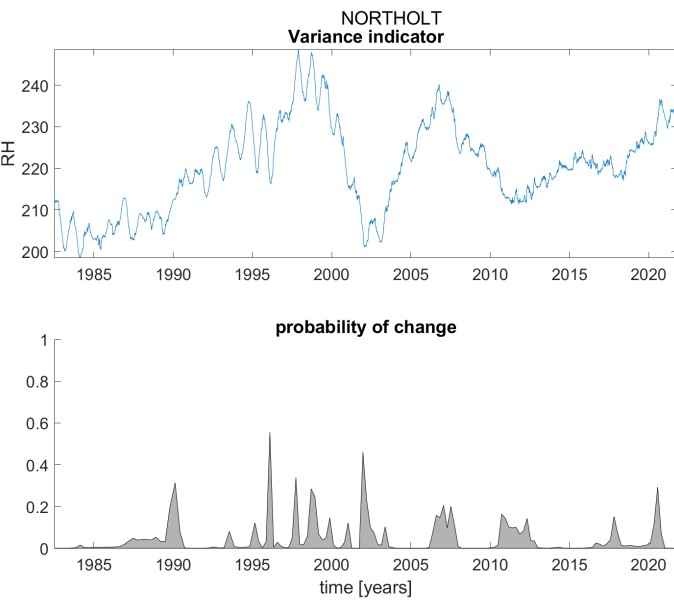

**Figure B17.** Variance indicator (upper panel) and probability of detection of changes in the variance indicator (lower panel) for station Northolt.

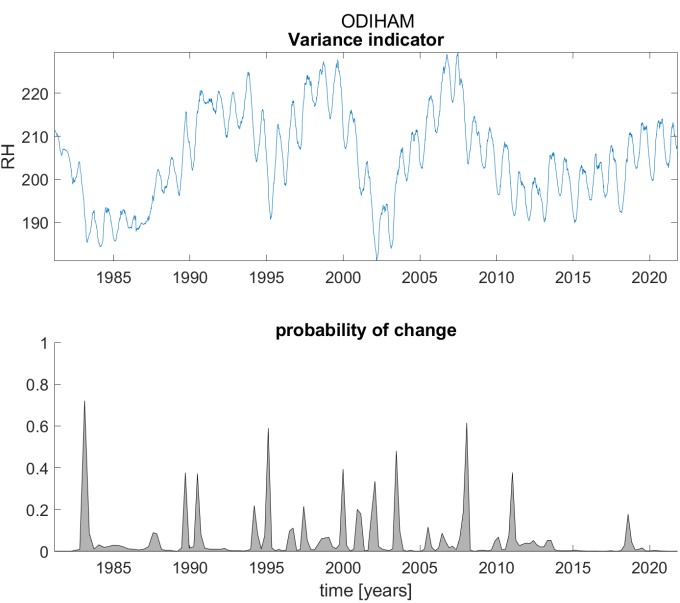

**Figure B18.** Variance indicator (upper panel) and probability of detection of changes in the variance indicator (lower panel) for station Odiham.

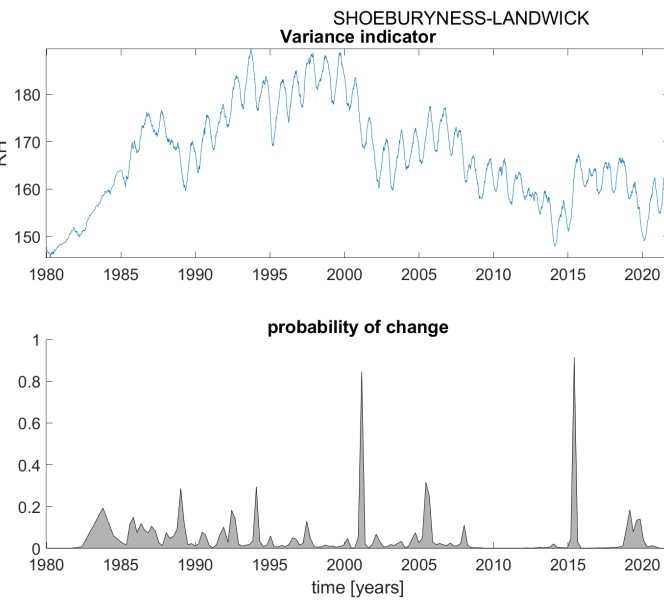

**Figure B19.** Variance indicator (upper panel) and probability of detection of changes in the variance indicator (lower panel) for station Shoeburyness-Landwick.

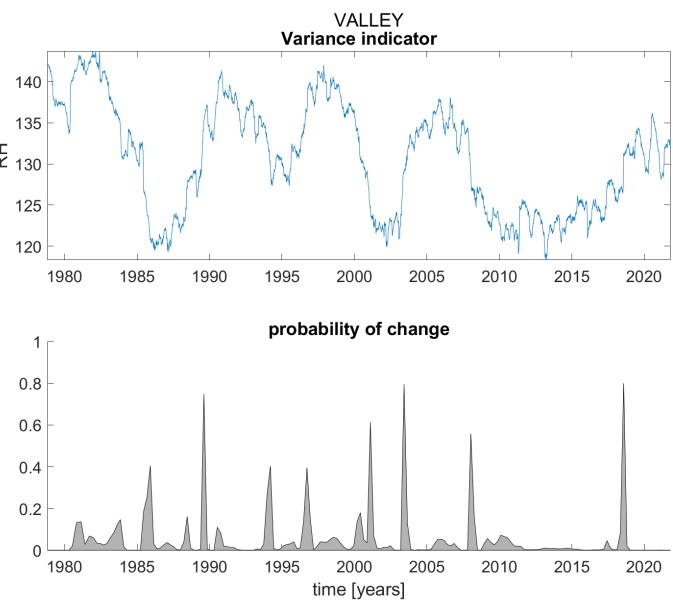

**Figure B20.** Variance indicator (upper panel) and probability of detection of changes in the variance indicator (lower panel) for station Valley.

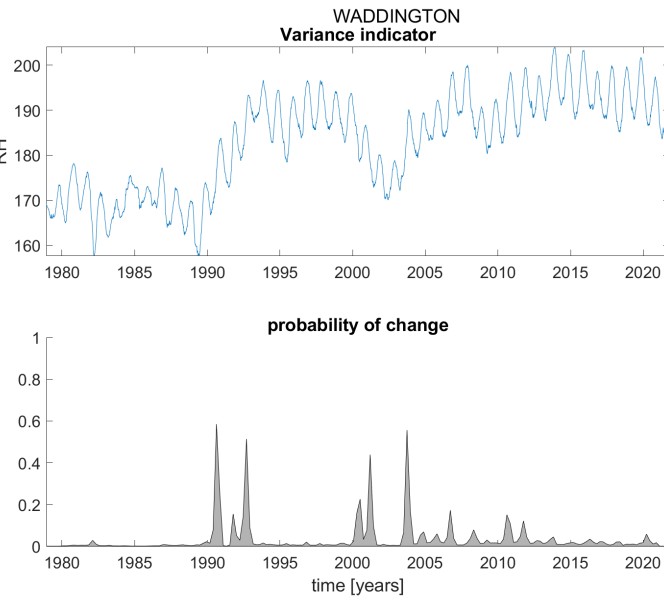

**Figure B21.** Variance indicator (upper panel) and probability of detection of changes in the variance indicator (lower panel) for station Waddington.

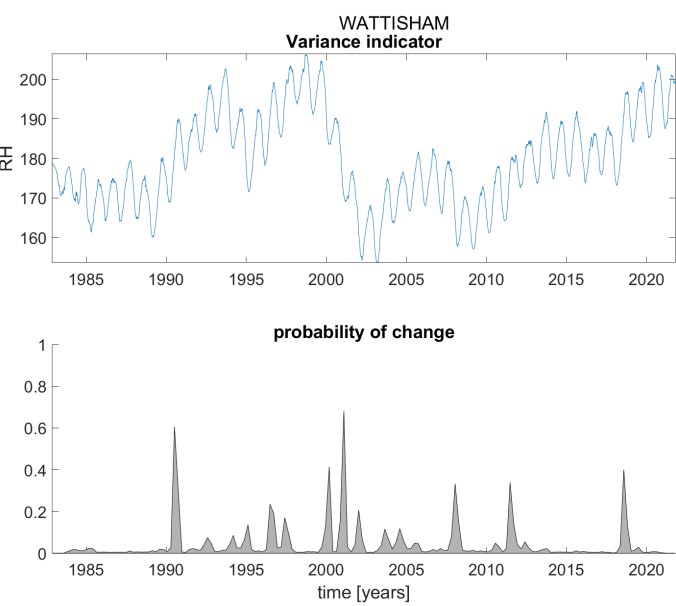

**Figure B22.** Variance indicator (upper panel) and probability of detection of changes in the variance indicator (lower panel) for station Wattisham.