# Peer review of "Tipping point analysis helps identify sensor phenomena in humidity data"

_EGUsphere, 2025_

## Author Comment (AC2)

**Response to Dr Chris Boulton (Reviewer#1)'s comments**

on paper "Tipping point analysis helps identify sensor phenomena in humidity data"

by *V. N. Livina, K. Willett, and S. Bell*

We would like to thank the Reviewer for the positive opinion about novelty of our approach.

**Comment 1.** *My main suggestion would be to include variance analysis too as an EWS.*

**Response.** We have included variance indicators in the appendix (figures B1-B22) and corresponding text.

**Comment 2.** *With an expected increase in the signal-to-noise ratio as newer sensors are included, the AR(1) should increase as observed in your work, but the variance would decrease at the same time. It seems that the combination of these would aid your work as it could rule out any "natural" change in the system itself. Smith et al. (2023) shows an example of this.*

**Response.** We have added the suggested reference. Accordingly, we added discussion of variance and autocorrelations and their suitability for sensor change detection.

**Comment 3.** *I think it's important to highlight how these EWS are affected by the changes in measurement circumstances, from the viewpoint of the other stages in tipping point analysis such as prediction and the chance of false positives.*

**Response.** We have added the following text in the Conclusion: "We have demonstrated that autocorrelations are both sensitive and robust in detection of known sensor changes. While not all metadata of measurement circumstances may be available, autocorrelation indicators provide the tool for scanning datasets for such changes, whose rapid development may help destinghuish them from the false positive signals of climatic changes. This should be taken into account in development of predictive techniques based on EWS signatures."

**Comment 4.** *I also had some slight confusion about how and why ERA5 data is used at all. It would be good to explain that the large gaps in data (I assume) come from the station data and not ERA5. Also, why can't the station data measurement just be used since the method used to create the reanalysis will not have the same issues?*

**Response.** We have added the following explanation: "While there were station-level measurements of pressure (variable 'stnlp'), these records were often short or patchy and the data quality was quite poor. Instead of these station pressure measurements, we used a climatological surface pressure from the nearest gridbox of the ECMWF ERA5 reanalysis product."

**Comment 5.** *Lines 77-80: This section is slightly confusing. I think it's suggesting that the AR(1) has to reach 1 as a critical value but Kendall's tau is also mentioned. I would be wary of saying that AR(1)=1 is critical when detrending has occurred in the time series it is calculated on as this alters the absolute value of AR(1). I would also refer to a "time series" of the indicator throughout rather than "curve".*

**Response.** We have reformulated this sentence as follows: "If the indicator rises to a critical value of 1 (as one is the maximal value of normalised autocorrelation, provided no detrending or filtering is applied to the input data), this is a signature of early warning signal of critical behaviour. However, when any pre-processing is applied to the input data, this is likely to reduce autocorrelation values, and indicator may not reach the maximal value 1. In this case, the important property of the indicator is its monotonic increasing trend. Such a monotonic trend of the EWS indicator can be estimated, for instance, using Kendall rank correlation."

We have also replaced "curve" with "time series".

**Comment 6.** *Lines 91-94: This section may not be needed. The potential plots here have not been estimated, for example.*

**Response.** Because we mentioned detection and forecasting in the overview of tipping point analysis in lines 67-69, we prefer to keep their explanation in lines 91-94. To address the comment, we have added the following sentence at the end of the paragraph: "In the current paper, we do not perform detection and forecasting stages of tipping point analysis and focus on anticipating tipping only EWS)."

**Comment 7.** *Line 113: What was wrong with the station that wasn't used?*

**Response.** Data quality (large gaps) was the main issue with the dataset. We mentioned this, but to clarify it better, we reformulated it like this: "After removing data before the largest gap in each of them, 55 stations were selected (the 56th being too short after truncation)."

**Comment 8.** *Fig. 1: What window length is being used here? I would also centre the x-axis on -1 to 1 in both panels.*

**Response.** We have adjusted the x-axis of the left panel of Fig.1. The size of the sliding window was actually mentioned in the caption of Fig.3, but the reviewer is right, it should be stated earlier. We have added in the caption of Fig.1: "The window size for calculation of indicators was 10% of time series length (as the records had different lengths)."

**Comment 9.** *Page 6: I think it's important to say what the actual window length was that was used, and personally I would suggest trying a longer window length as well*

*to see how the results contrast given the discussion on this page. Also, does the BCP analysis require any a priori input on the change of form that is searched for (e.g. looking for a certain number of changepoints)? If so, this should be stated.*

**Response.** 10% of the dataset length is a compromise of sensitivity of indicators and sufficient aggregation of data within a window. Because we are interested in instrumental changes, the size of the window cannot be taken too large, as data aggregation in a large window would suppress the signal of such changes, and moreover, a large window would increase the uncertainty in timing the event of the instrument change.

We did not use any asumption on the number of change points, the BCP technique has no such parameter.

**Comment 10.** *Fig. 2: I feel like this could be better represented by a continuous blue to red scale rather than the size of the circles as I find it hard to distinguish between the sizes (except the blue ones look small).*

**Response.** We have replaced Fig.2 with a more clear version, with updated explanation in the caption.

**Comment 11.** *Fig. 3: The red box is not defined in the figure caption.*

**Response.** We have added the following explanation in the caption: "The red boxes denote the intervals of transitions: in the input time series the changes are not visible, whereas in the detrended series one can notice the change of pattern, which is then clearly detected by the EWS indicator."

**Comment 12.** *Line 180: There are no detections in the 1980s in Figures 7 or 8.*

**Response.** We have removed "1980s" in this line.

**Comment 13.** *Fig. 6-8: It would be good to add the red crosses on the bottom panels each time to see how they match up more clearly.*

**Response.** We have modified these three figures and the captions accordingly.

**Comment 14.** *Line 191: The shifts in the 1980s happen in the Appendix but not in the figures in the main paper.*

**Response.** We added the comment in brackets: "(see figures in the main text and in the Appendix)".

**Comment 15.** *Appendix Table: Is this for detections above 0.8 with the BCP analysis? If so, it should say in the caption.*

**Response.** We have added this in the caption.

We hope that the revised manuscript is now suitable for publication in the "Geoscientific Instrumentation, Methods and Data Systems" magazine.

Yours sincerely,
V. N. Livina, K. Willett, and S. Bell

---

## Author Comment (AC4)

**Response to Reviewer#2's comments**
on paper "Tipping point analysis helps identify sensor phenomena in humidity data"
by *V. N. Livina, K. Willett, and S. Bell*

We would like to thank the Reviewer for the positive opinion about novelty of our approach.

**Comment 1.** *The authors state that the uncertainty in the timing of breakpoint detection using EWS methods is "on the order of several weeks". This statement is too vague and should be quantified more precisely. Ideally, the uncertainty should be expressed as a time interval, with a more detailed explanation of its origin. For instance, spectral analysis or low-rank reconstructions such as Singular Spectrum Analysis (SSA) may introduce smoothing effects that act as data-driven low-pass filters. When high-frequency noise is removed, residuals may show inflated autocorrelation, which could bias diagnostics based on autocorrelation patterns, such as EWS and breakpoint detection. This aspect should be discussed explicitly.*

**Response.** Because the series vary in length, while we are using the window length of 10% of the series length, the sliding window aggregates data of different length in each case. They are comparable but vary from series to series. Furthermore, there is an uncertainty due to aggregateion of data within a selected window. This uncertainty can be reduced by using smaller and smaller window, if there is a task of precise timing of the change. In this paper, we aim to demonstrate that such detection is possible in principle, and in further work we can consider the problem of timing with reduced uncertainties.

The reviewer is correct that pre-processing of data may influence autocorrelations: when a smoothing filter is applied, correlations increase, which may lead to very high level of fluctuations of autocorrelation function close to critical value 1. Such EWS values are usually not informative, and detection of sensor changes would be unfeasible. However, in our paper we apply detrending, which acts in the opposite way to smoothing: detrending removes the low-frequency trend, and autocorrelations diminish.

Despite these modifications, the main object of interest is the monotonic trend in the EWS indicator, which denotes the change in the dynamics of fluctuations. The absolute values of the EWS indicator are less important in this context.

We have added these comments into the paper.

**Comment 2.** *The authors focus exclusively on relative humidity (RH) time series. While RH may be more stationary than temperature and is indeed relevant for the surface energy balance, it is rarely used as a primary indicator in global warming*

*assessments. Temperature, in contrast, is the most studied, standardized, and policy-relevant climate variable. Although it is important to explore all surface variables to fully understand climate change, the omission of temperature series limits the broader applicability and scientific impact of the study. The rationale for this choice should be better justified. Ideally, the method should also be tested or validated on temperature data, which are better documented and more commonly used in homogenization and climate trend studies.*

**Response.** The idea of the paper appeared specifically in the context of the analysis of relative humidity, because the MetOffice datasets HadISD required investigation in the context of sensor changes. The paper has addressed the request from industry, hence the choice of the variable. We agree that other variables can represent further interesting case studies, and we plan to investigate this in future work.

**Comment 3.** *RH measurements over the decades have been affected not only by changes in the mean but also in their variability, particularly before 1990. These changes may result from sensor issues such as limited sensitivity or response time. The authors should explain how such biases might influence the performance of EWS methods in detecting breakpoints, especially since EWS methods may rely on changes in variance and autocorrelation.*

**Response.** The reviewer is right that EWS indicators are based on dynamics of autocorrelations. Historic measurements may contain information of changes not only from analogue-digital transformations but also changes of older analogue instruments along the course of time (replacement after malfunction and the like). The EWS indicators will detect such changes as well, and in principle these cannot be treated as biases but rather be interpreted as historic measurement changes.

**Comment 4.** *To evaluate the added value of the proposed EWS technique, it would be useful to compare it with more established methods, such as Observation-minus-Background (O-B) diagnostics from atmospheric reanalyses, at least for the two cases discussed in the main text. Comparison with widely used methods for detecting breakpoints and assessing the quality of time series may be beneficial to highlight the quality of the proposed approach. Such a comparison would help clarify why some breakpoints remain undetected by EWS and highlight the potential advantages or limitations of the approach compared to other techniques.*

**Response.** The observation-minus-background method is discussed in the added literature. The essence of the approach is estimation of error covariance, which then can be used for analysis of observational deviations. However, in such method, several assumptions are applied, such as absence of correlations between observation and background errors, and the analysis errors are expected to be related linerally

with observation and background errors. We note that such assumptions are not imposed when applying the technique proposed in the current manuscript.

We have added this text into the manuscript.

In addition, we tested three previously published changepoint techniques on Bingley dataset. These three did not provide satisfactory results; we did not include them in the manuscript but provide the results here to address the reviewer's concern:

– **Wavelet Changepoint Detection.** For the Bingley dataset, whose length is 263763 points, the algorithm requested a 263763x263763 array allocation of size 518.3GB and crashed the calculations.

– **Canonical changepoint detection** (Killick et al, Journal of the American Statistical Association, 107 (500), 2012). The algorithm requires a parameter of maximum possible changepoints, which we assigned as 10. The results, indeed, provided 10 detected points for the Bingley dataset, but none of those coincided with the change from analogue to digital in the 1990s, as shown in the following figure.

[Figure]

Figure 1: Bingley dataset analysed with canonical changepoint detection algorithm, assuming 10 change points as a parameter. Vertical green lines denote ten detected change points.

– **Bayesian change point detection**
(Kaiguang, https://github.com/zhaokg/Rbeast/releases/tag/1.1.2.60)
This technique, again, detected several change points, but they did not match the MetOffice metadata records as was detected by the tipping point technique - see the following figure.

Computationally, our technique is much lighter; its results are more accurate as illustrated in the humidity data; it is easier to apply and interpret its results.

[Figure]

Figure 2: Bingley dataset analysed with Bayesian changepoint detection algorithm. Detected changes are denoted by steps in 2nd panel and changes of colour in 3rd panel.

**Comment 5.** *The exclusion of time series with unresolved breaks removes a substantial portion of potentially valuable climate information. The authors should take a clearer position on this issue: is the exclusion of parts of the time series with the largest temporal gaps recommended as a general strategy, or should these series be reconstructed through infilling or correction techniques? The manuscript would benefit from a more explicit discussion of this trade-off.*

**Response.** We have removed large gaps, i.e., intervals with absent data - these contain no statistics and it is not clear how to use them for EWS analysis. Filling such gaps would be possible only with highly hypothetical artificial data, which would affect autocorrelations and distort EWS estimation. Truncating intervals with large gaps is the simplest solution, but we may consider alternatives in future work.

We have added this text to the paper.

**Comment 6.** *Line 117-120: Is temperature not recorded at the stations alongside RH? If so, it is not fully clear why both pressure and dew point ERA5 data are needed here, especially if they are not being used to fill data gaps. Please clarify this point.*

**Response.** We have added the following text: "While there were station-level measurements of pressure (variable 'stnlp'), these records were often short or patchy and the data quality was quite poor. Instead of these station pressure measurements, we used a climatological surface pressure from the nearest gridbox of the ECMWF ERA5 reanalysis product."

Temperature and dewpoint variables were from the HadIST dataset provided by the MetOffice.

**Comment 7.** *Line 124-125: Provide additional context on the use of Singular Spectrum Analysis to the reader. Explain how and why it was applied in this specific case.*

**Response.** We have added the following text: "SSA decomposes the input time series into a sum of components (low-frequency trend, seasonal modulations, and detrended noise. The method is based on the singular value decomposition of a special matrix constructed from the time series. The advantage of the method is that neither a parametric model nor stationarity conditions are to be assumed for the time series. This makes SSA a versatile tool of time series analysis."

**Comment 8.** *Lines 143-145 vs. 182-184: The rationale for choosing short time windows to avoid conflating long-term changes with breakpoints appears inconsistent with later claims that long-term trends are still detectable. Please clarify the limitations and implications of the chosen window size.*

**Response.** We have added the following explanation: "EWS indicators have sensitivity for detection of short-term trends. However, such an indicator can still contain signatures of longer-term trends, although with noisy patterns. The choice of the window is requires balancing of the sampling rate of time series and the dynamics of the phenomenon of interest. For example, for studying climate change effects, which manifest at the scale of several decades, it is necessary to consider an indicator with sufficiently large sliding window; for studying rapid changes in the time series, which may be due to instrument change or deterioration, much smaller window size is necessary. Indicators with large window size usually have smoother patterns due to aggregation and average of large subsets of data."

**Comment 9.** *Line 162-163: This section offers an excellent opportunity to demonstrate the utility of the EWS approach by comparing detected breakpoints with available metadata. Consider including a table (even in the appendix) with station metadata and detected breaks, and comment on the comparison of the breakpoint detection dates.*

**Response.** Figs.6-8 contain red star markers that are based on the metadata of known changes recorded by MetOffice. We also updated these figures with markers in the bottom panels for better clarity.

**Comment 10.** *Line 167: It would be helpful to explain how the levels of autocorrelation and the resulting probabilities of breakpoint detection can vary depending on the nature and magnitude of the underlying break. For instance, abrupt shifts in the mean versus gradual drifts or changes in variance may produce different statistical signals, leading to varying detection sensitivity. Moreover, spectral smoothing or reconstruction methods (such as SSA) may affect the autocorrelation. These aspects should be discussed in detail to justify the robustness of the detection criteria.*

**Response.** In principle, abrupt shifts, which are less subtle than changes in pattern in a time series, can be often detected by eye or using simple statistical techniques like comparison of mean values.

We have added the following text: "The SSA technique was applied to the input data to obtain detrended fluctuations. Such detrending can affect the level of autocorrelations but not the trend in the indicator time series. EWS is estimated as monotonous positive trend, and this trend is present in both indicators of the raw and detrended data."

**Comment 11.** *Lines 168-170: Breakpoints may arise not only from instrumentation changes but also from relocations, calibration issues, or sensor malfunctions. This should be described in more detail.*

**Response.** We have added the following text: "We did not attempt to distinguish different types of sensor issues that the reviewer listed above. Rather, we wanted to demonstrate that general detection is possible, which can then guide further investigation of such issues."

**Comment 12.** *Lines 171-172 (Figure 6-7): Bingley and Camborne stations reportedly show no documented changes before 1990. Is this because there were no changes, or are the metadata simply unavailable? This is important information for readers and should be clarified.*

**Response.** One of the motivations for this study was that in many cases of long-term climatological observations, metadata might be missing or difficult to obtain (some documentation may still be not digitalised).

We have added this text to the manuscript.

**Comment 13.** *Lines 175-179: See general comment above regarding comparisons with other break detection methods.*

**Response.** We have added extensive literature review and references to compare the proposed technique with other methods.

**Comment 14.** *Please consider expanding the bibliography to include more contributions from the broader climate science community on break detection, including methods based on metadata, O-B fields, Bayesian approaches, etc.*

**Response.** We have added references and discussion on several other methods.

We hope that the revised manuscript is now suitable for publication in the "Geoscientific Instrumentation, Methods and Data Systems" magazine.

Yours sincerely,
V. N. Livina, K. Willett, and S. Bell